# Diverse Cretaceous larvae reveal the evolutionary and behavioural history of antlions and lacewings

Davide Badano [1], Michael S. Engel [2,3], Andrea Basso [4], Bo Wang [5,6] & Pierfilippo Cerretti [7]

Myrmeleontiformia are an ancient group of lacewing insects characterized by predatory larvae with unusual morphologies and behaviours. Mostly soil dwellers with a soft cuticle, their larvae fossilize only as amber inclusions, and thus their fossil record is remarkably sparse. Here, we document a disparate assemblage of myrmeleontiform larvae from the mid-Cretaceous amber (99 Ma) of Myanmar, evidence of a considerable diversification. Our cladistic analysis integrating extant and extinct taxa resolves the fossils as both stem- and crown-groups. Similarities between extinct and extant species permit inferences of larval ethology of the fossil species through statistical correlation analyses with high support, implying that morphological disparity matched behavioural diversity. An improved understanding of the evolutionary history of antlions and relatives supports the conclusion that hunting strategies, such as camouflage and fossoriality, were acquired early within the lineage.

---

[1] Dipartimento di Scienze della Terra, dell'Ambiente e della Vita, Università degli studi di Genova, Corso Europa 26, 16132 Genoa, Italy. [2] Division of Entomology, Natural History Museum, and Department of Ecology & Evolutionary Biology, University of Kansas, Lawrence, KS 66045, USA. [3] Division of Invertebrate Zoology, American Museum of Natural History, New York, NY 10024, USA. [4] Laboratory of Genomics, DAFNAE, Agripolis–University of Padova, Viale dell'Università 16, 35020 Legnaro, Italy. [5] State Key Laboratory of Palaeobiology and Stratigraphy, Nanjing Institute of Geology and Palaeontology and Center for Excellence in Life and Paleoenvironment, Chinese Academy of Sciences, 39 East Beijing Road, Nanjing 210008, China. [6] Key Laboratory of Zoological Systematics and Evolution, Institute of Zoology, Chinese Academy of Sciences, Beijing 100101, China. [7] Dipartimento di Biologia e Biotecnologie "Charles Darwin", Sapienza Università di Roma, Piazzale A. Moro 5, 00185 Rome, Italy. These authors contributed equally: Davide Badano, Michael S. Engel. Correspondence and requests for materials should be addressed to D.B. (email: davide.badano@gmail.com) or to B.W. (email: bowang@nigpas.ac.cn) or to P.C. (email: pierfilippo.cerretti@uniroma1.it)

The immature stages of insects embody a wealth of biological information, particularly so for holometabolous insects whereby the larval stages can often lead dramatically different lives from those of their corresponding adults[1–4]. Data from larvae are vital in everything from effective biological control programs to the elucidation of major evolutionary patterns[5]. Despite the importance of larvae for understanding evolutionary relationships and biological phenomena among insects[6,7], the fossil record of insect larvae remains a largely untapped resource. This is the result of several factors such as difficulty in properly identifying particular larval forms, distinguishing among stages, and often poor preservation owing to their frequently soft integument. Nonetheless, fossil larvae, when studied properly, have proven to be a rich source of information for evolutionary studies[8,9]. The preservation of larvae in amber, whereby their finer features are preserved with greater fidelity than via other modes of fossilization, allows for life-like comparison between extinct and extant larval forms.

The lacewings, antlions, owlflies, and their relatives comprise the insect order Neuroptera, and for more than 150 years the study of lacewing larvae has proven critical to understanding broad evolutionary patterns across the clade[10]. In addition, not only have larvae been vital to resolving relationships among neuropteran lineages, but these larvae also exhibit unique and remarkable biologies. Neuropteran larvae are famed for their various modes of predatory behaviour, using in some cases exogenous materials as camouflage for approaching prey and avoiding their own predators, to excavating elaborate funnels in which to entrap surface-dwelling victims, as is famous among antlions[11] (Fig. 1). Neuroptera are also an ancient lineage, with many clades extending back to the Triassic and Jurassic, and the group as a whole is known from as far back as the Permian[5,12,13]. Neuroptera, together with Megaloptera and Raphidioptera, form the superorder Neuropterida, in turn sister to Coleoptera +Strepsiptera[12] (Fig. 2). Among this diversity, the clade Myrmeleontiformia, comprising the superfamilies Psychopsoidea and Myrmeleontoidea[13], includes some of the largest and most diverse of all lacewings, particularly among antlions, Myrmeleontidae. Myrmeleontiform lacewings are among the most characteristic insects of arid environments worldwide.

Although myrmeleontiform adults are usually predators, like their larvae, they are of delicate appearance, varying from dragonfly-like to butterfly-like species, with massive and often conspicuously marked wings (Fig. 1a–c). Their campodeiform larvae are voracious ambush predators, with massive trap jaws and sometimes with bizarre body types, including those with elongate, slender necks or bodies covered in "spiny" or branched tubercles (Fig. 1d–f). Moreover, these insects are characterized by a wide range of ecological adaptations, spanning from arboreal to deep-soil dwellers[11] (Fig. 1g, h). Such morphological and ecological disparity is remarkable in a relatively small group (at least by insect standards), with only about 2340 extant species. Recent phylogenomic analyses and time-calibrated phylogenies have suggested that the stem-group of Myrmeleontiformia perhaps originated in the Permian[12,14,15]. The earliest definitive fossils of Myrmeleontiformia are from the Late Triassic, and by the Jurassic and Cretaceous this clade was relatively diverse[5,13,16–18]. Interestingly, the Cenozoic record of myrmeleontiform lacewings is comparatively poor[5,19].

Generally, fossil larvae of Myrmeleontiformia are scarce and unevenly distributed across time and space, with amber deposits representing rare "windows" to their past diversity and morphology[20–23]. An extraordinary collection of larvae of Cretaceous Myrmeleontiformia has been recently found in the rich amber deposits of Myanmar (dated at $98.8 \pm 0.6$ Ma[24]). These deposits encompass one of the most diverse Mesozoic amber biotas

presently known[25,26]. Adult Myrmeleontiformia from Burmese amber have been rather extensively studied[5,22,27–30]. By contrast, although larvae are also numerous in Burmese amber, these species have only been briefly explored in terms of their camouflaging behaviour, similar to that performed by extant species[15].

A rich assemblage of immature Myrmeleontiformia from Burmese amber allows us to test the importance of extinct phenotypes in estimating the phylogeny and evolution of larval traits, as well as in statistically correlating morphology with given behavioural traits. Our reconstruction suggests that fossorial specializations evolved more than once across myrmeleontiforms from arboreal ancestors. Moreover, we find strong correlations between a selection of morphological traits and two hunting strategies of these ambush predators—camouflaging and fossoriality—allowing us to reconstruct habits of extinct species.

## Results
### Systematic palaeontology

Order Neuroptera Linnaeus, 1758

Stem-group Myrmeleontiformia

*Macleodiella electrina* Badano & Engel gen. et sp. nov. (Fig. 3a, Supplementary Fig. 1a–c)
LSID (Life Science Identifier): urn:lsid:zoobank.org:act:B2B9C3F0-1341-47D2-AAFB-A8F57D16D0BA; urn:lsid:zoobank.org:act:3282500D-0015-4F2E-A0F8-FD6FFEDF89E9.

**Type species**. *Macleodiella electrina* Badano & Engel sp. nov.
**Etymology**. The generic name honours Ellis G. MacLeod (1928–1997) for his outstanding studies on larval Neuroptera. The noun is feminine. The specific epithet is an adjective of Latin derivation, meaning "of amber".
**Type material**. Holotype: AMNH JCZ-Bu30 (Fig. 3a, Supplementary Fig. 1a–c), excellent state of preservation, dorsal view only. Probably 1st instar.
**Description (larva)**. Measurements (Supplementary Table 1). Head capsule subrectangular in shape, much longer than wide, reaching maximum width in proximity of ocular area and gently tapering posteriorly. Lateral side of head with seven sessile stemmata. Clypeolabrum covering whole anterior margin between mandibles. Anterior labral margin projecting forward as median paired triangular processes, each bearing a short seta at apex; internal process smaller than external one. Anterior tentorial pits hardly noticeable, running obliquely from base of antennae to median portion of head capsule. Posterior tentorial grooves on anterior third of head capsule. Gular sclerite triangular, relatively small, less than 1/5th of head length. Hypostomal bridge as a thin line running along entire ventral surface of head. Temple as a slight constriction of posterior margin of head capsule. Antenna slender, much longer than mandible, composed of three elongate flagellomeres. Scape wider than following antennomeres, subconical in shape; pedicel elongate; 1st flagellomere 1.5 times longer than pedicel, 2nd flagellomere slightly longer than pedicel, 3rd flagellomere 1/5th of 2nd flagellomere, tapered apically and bearing an apical seta. Mandible relatively slender and strongly curved inward, as long as head capsule, widely separated at base. Mandible provided with three thin equidistant teeth, subequal in size and much longer than its width. At least one seta present between mandibular base and basal tooth. Two or three pseudoteeth interspersed with short setae present between basal and median teeth. Portion of mandible between median and apical teeth bearing only a few setae or none. External margin of mandible without setae. Labial

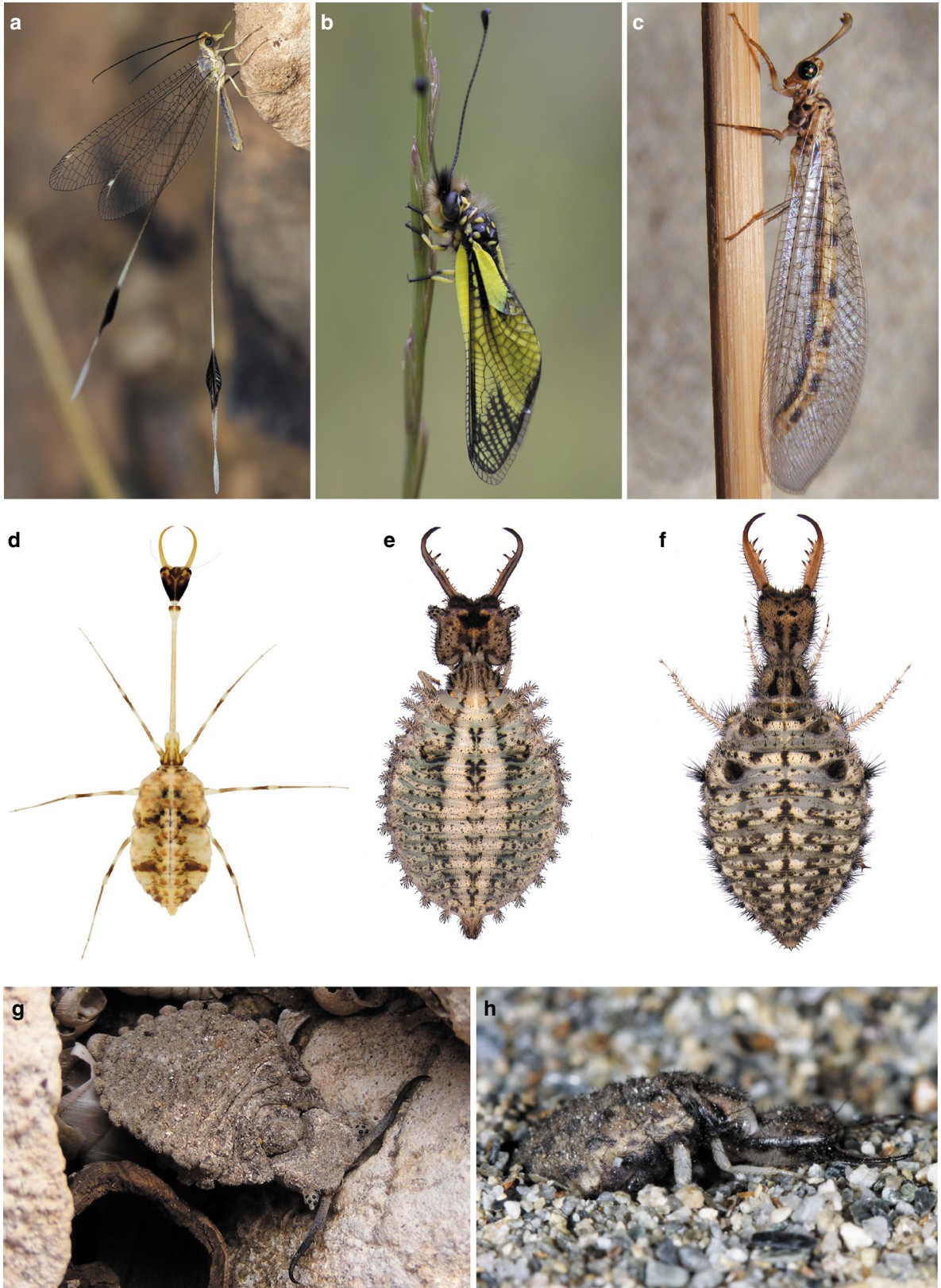

**Fig. 1** Diversity and biology of extant Myrmeleontiformia. **a** *Halter halteratus* (Forskål), Nemopteridae. **b** *Libelloides latinus* (Lefèbvre), Ascalaphidae. **c** *Myrmeleon fasciatus* (Navás), Myrmeleontidae. **d** Larva of *Pterocroce capillaris* (Klug), Nemopteridae. **e** Larva of *Libelloides latinus* (Lefèbvre), Ascalaphidae. **f** Larva of *Macroleon quinquemaculatus* (Hagen), Myrmeleontidae. **g** Camouflaging larva of *Bubopsis agrionoides* (Rambur), Ascalaphidae. **h** Fossorial larva of *Creoleon lugdunensis* (Villiers), Myrmeleontidae. Photos credits: **a** by Cosmin O. Manci; **b**, **h** by Claudio Labriola; **c**–**g** by Davide Badano

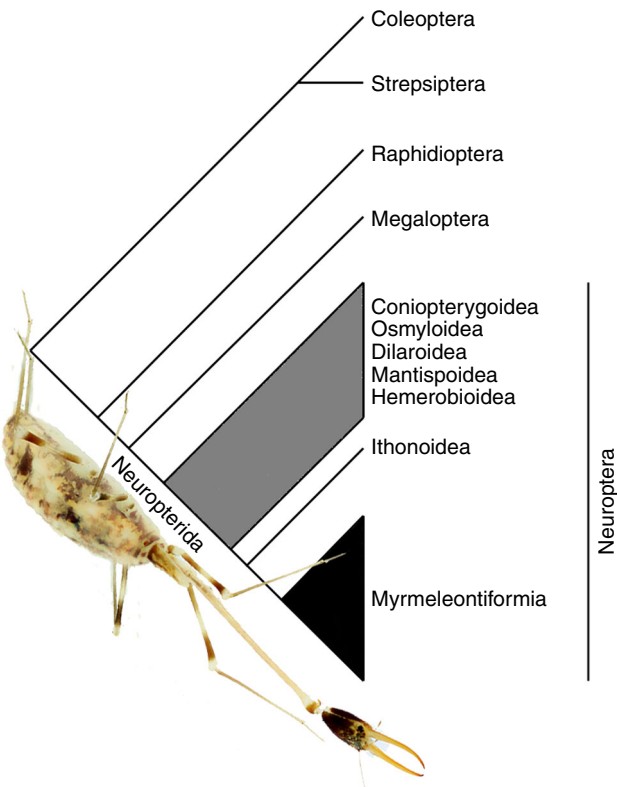

**Fig. 2** Simplified cladogram of the internal relationships of Neuropteroidea. The larva of the nemopterid *Pterocroce capillaris* (Klug) highlights Neuropterida

palpus elongate, as long as 2/3rd of head capsule, composed of three palpomeres. Apical palpomere elongate, strongly tapered apically. Palpiger similar in shape and size to palpomeres. Head capsule smooth, covered with sparse, thin setae. Neck region well defined but relatively short. Cervix well recognizable as a membranous, collar-like area between posterior, neck-like region of head capsule and prothorax. Prothorax strongly sclerotized, much longer than wide and noticeably narrower than both head capsule and remainder of body; separated from mesothorax by membranous subsegments. Meso- and metathorax membranous. Mesothoracic spiracle sessile. Cuticular insertions of leg muscles recognizable as small, dorsal, sclerotized plates. Legs relatively short and robust, of similar size. Femora and tibiae covered with a few setae, distal portion of tibiae with a pair of robust setae; pretarsi equipped with inconspicuous claws and a trumpet-shaped arolium. Abdomen composed of nine segments, with a blunt and wide apex. Pleurae swollen externally and slightly raised dorsally. Body covered with sparse, thin setae.

**Comments**. *M. electrina* has a unique combination of character states allowing for it to easily be distinguished from any other known larva of Neuroptera. This species is characterized by the presence of spiniform processes on the clypeolabrum and the abdominal segment 9 wider than long. *Macleodiella* is easily recognizable by having the head capsule much longer than wide, the antenna as long as the mandible but with few flagellomeres, a relatively thin mandible with acute teeth much longer than the mandibular width, pseudoteeth present, the body greatly elongate and without processes, the pronotum sclerotized and much longer than wide, and arolia present.

*Cladofer huangi* Badano, Engel & Wang gen. et sp. nov. (Supplementary Fig. 1d, e)

LSID: urn:lsid:zoobank.org:act:DBB89A2E-7921-46C1-84CC-94F08E788C4D; urn:lsid:zoobank.org:act:06142ED1-80FE-4F5C-8A64-2ACFBF44459E.

**Type species**. *Cladofer huangi* Badano, Engel & Wang sp. nov.
**Etymology**. The generic name derives from Greek, meaning "branch-bearer" and should be treated as masculine. The species is dedicated to Huang Yijen, who brought it to the attention of the authors.
**Type material**. Holotype: NIGP152466 (Supplementary Fig. 1d, e), partly decomposed, in an opaque amber piece.
**Description (larva)**. Measurements (Supplementary Table 1). Head capsule as long as wide, slightly tapered posteriorly. Temple area smooth, not pronounced. Stemmata sessile, grouped on a short bulge. Anterior labral margin concave. Antenna slender, much longer than mandible, with numerous (>30) flagellomeres. Distal margin of each flagellomere with thin bristles. Mandible much longer than head capsule, relatively slender and straight, curved inward, with seven teeth, which are progressively longer toward apex. Mandibular teeth interspersed with pseudoteeth and thin setae. External margin of mandible covered with short, thin setae. Labial palpus thin, as long as two-third of mandible, apical palpomere elongate. Posterior portion of head with a short neck region. Prothorax slightly longer than wide, without setiferous processes. Meso- and metathorax membranous, each equipped with one extremely long, thin setiferous process on dorsolateral surface of body. Thoracic processes with long setae arranged as shifted by half an interval. Legs slender. Composed of at least nine recognizable segments. Abdominal segments 1–7 with one dorsal and one ventral series of setiferous processes; processes of dorsal series arranged dorsally, similar in shape, length, and chaetotaxy to those present on thorax; processes of ventral series scolus-like, but relatively short and stout. Abdominal segment 8 without processes. Abdominal segment 9 longer than wide.

**Comments**. Although the holotype is poorly preserved, the morphology of *C. huangi* is remarkably different from any other larva of Neuroptera. The presence of seven mandibular teeth is a remarkable diagnostic character of this species. *Cladofer* is characterized by one pair of meso- and metathoracic setiferous processes and conspicuous abdominal setiferous processes. The number and arrangement of setiferous processes on the body, with a single process on the thoracic segments and abdominal segment 8 devoid of protuberances, closely resembles the condition observed in Nymphidae. However, *C. huangi* is easily set apart from Nymphidae owing to the antenna as long as mandible with numerous flagellomeres, presence of multiple mandibular teeth, and an elongate body. The shape of the thoracic setiferous processes, being longer than the body width and comparatively thin, with alternately arranged setae, suggests that the similarities between *Cladofer* and Nymphidae are the result of convergence. Also of note is the close resemblance between the shape of the setiferous processes of *Cladofer* and those of unrelated Mesozoic stem-chrysopids.

Family Psychopsidae Handlirsch, 1906
*Acanthopsychops triaina* Badano & Engel gen. et sp. nov. (Supplementary Fig. 1f, g)
LSID: urn:lsid:zoobank.org:act:0323530A-94F2-464C-B6D9-6A572C512381; urn:lsid:zoobank.org:act:A9B11AFE-7281-4A3A-AD8F-77594316EEC2.

**Type species**. *Acanthopsychops triaina* Badano & Engel sp. nov.
**Etymology**. The generic, feminine, epithet is a composite name from Greek, with a prefix meaning "spine" and a suffix meaning

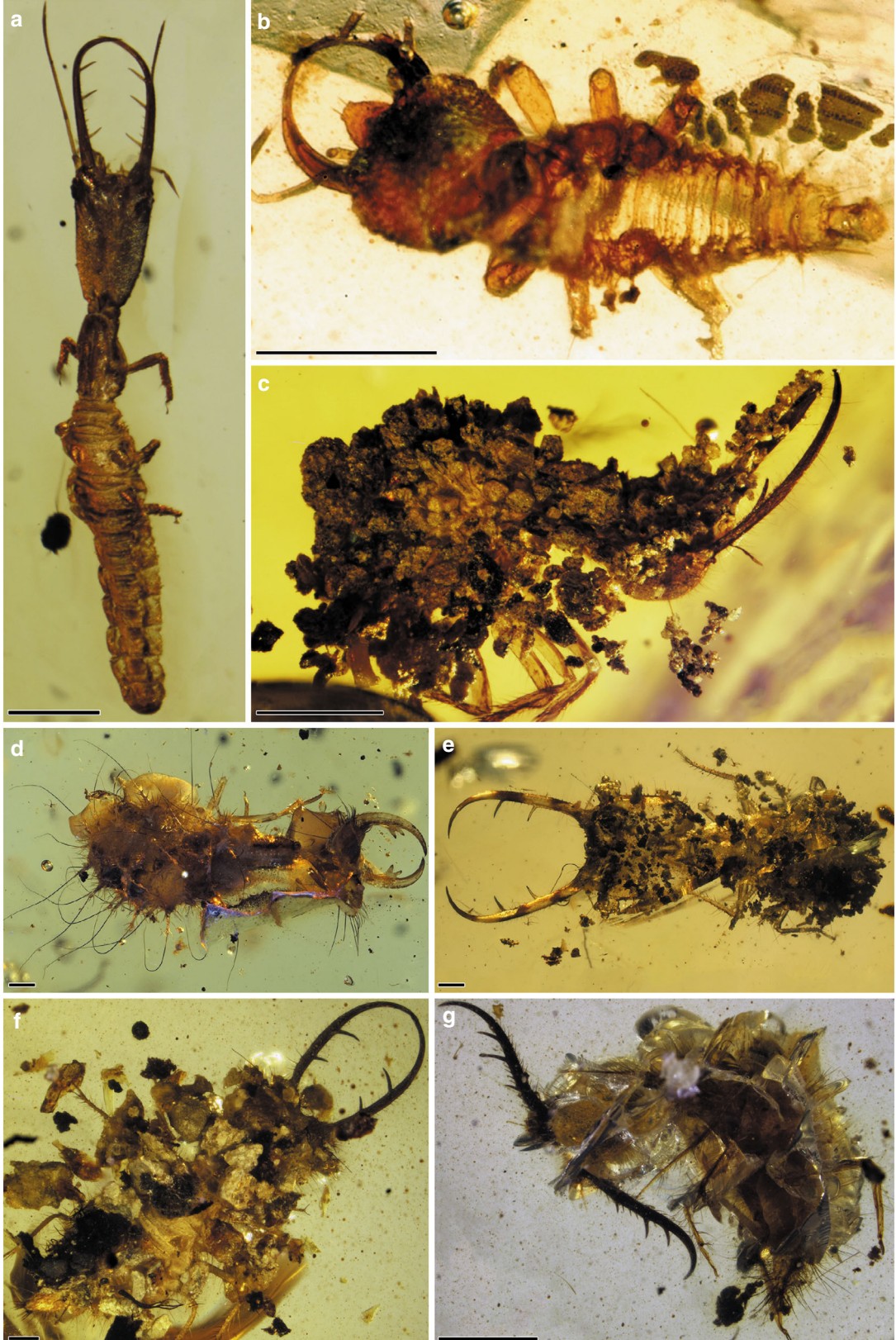

**Fig. 3** Diversity of the larvae of Myrmeleontiformia in mid-Cretaceous Burmese amber. **a** *Macleodiella electrina* gen. et sp. nov.,
holotype. **b** *Aphthartopsychops scutatus* gen. et sp. nov., holotype. **c** *Nymphavus progenitor* gen. et sp. nov., paratype. **d** *Electrocaptivus xui* gen. et sp. nov.,
holotype. **e** *Diodontognathus papillatus* gen. et sp. nov., holotype. **f** *Adelpholeon lithophorus* gen. et sp. nov., holotype. **g** *Pristinofossor rictus* gen. et sp. nov.,
holotype. Scale bars: 0.5 mm

"mind" or "soul", commonly used for silky lacewings. The specific epithet is a Greek name meaning "trident".

**Type material**. Holotype: AMNH JCZ Bu-404 (Supplementary Fig. 1f, g). Head well preserved but most of the thorax and abdomen were lost, with the exception of the legs. Presumably 1st instar.

**Description (larva)**. Measurements (Supplementary Table 1). Head capsule subrectangular in shape, longer than wide, slightly tapered posteriorly. Maximum width of head in proximity of ocular area. Margin of labrum with a large median triangular rostrum and smaller triangular processes laterally. Stemmata sessile. Antenna slender, much longer than mandible, composed of more than 20 flagellomeres. Mandible strongly sclerotized, sickle-shaped and evenly curved inward, longer than head capsule. Surface of head capsule with raised setae arising on papillae. Neck region not abruptly differentiated from posterior portion of head capsule. Cervical area collar-like, membranous. Legs relatively short and robust. Femora swollen, tibiae elongate; tarsi relatively short; pretarsal claws short.

**Comments**. This specimen is readily recognizable as a larva of Psychopsidae, despite a generally poor state of preservation, in having papillae on the head capsule; a relatively long, filiform antenna composed of numerous flagellomeres; a toothless mandible; a labrum with prominent rostrum; and a distinct neck region. This larva differs from all other known extant and fossil larvae of Psychopsidae in the highly characteristic trident-like shape of the anterior margin of the labrum, while in the other larvae of this family the rostrum is a single, subtriangular projection. The shape of the head capsule, gently tapering posteriorly, is also diagnostic of this species.

*Aphthartopsychops scutatus* Badano & Engel gen. et sp. nov. (Fig. 3b)
LSID:     urn:lsid:zoobank.org:act:D2567357-BDE5-4637-92DD-7A5CD16179E0; urn:lsid:zoobank.org:act:35CC68CC-2984-4950-8C9B-36A3C3397D5E.

**Type species**. *Aphthartopsychops scutatus* Badano & Engel sp. nov.

**Etymology**. The genus name derives from Greek, with a prefix meaning "undecaying, uncorruptible" and the common suffix for silky lacewings, meaning "mind" or "soul" and it is feminine. The specific epithet, from Latin, refers to the shield-like pronotum.

**Type material**. Holotype: AMNH JCZ-Bu197 (Fig. 3b). Presumably 1st instar.

**Description (larva)**. Measurements (Supplementary Table 1). Head capsule broad, slightly wider than long, slightly tapered posteriorly, i.e., maximum width of head capsule in proximity of ocular area. Margin of labrum with a relatively large median trapezoidal rostrum (1/2 of mandible length) with two erect paramedian setae at apex. Five sessile stemmata placed near antennal base. Antenna slender, much longer than mandible, composed of four flagellomeres. Scape slightly longer than wide; pedicel elongate, much longer than wide and longer than flagellomeres 1–3 combined; flagellomeres 1–3 slightly longer than wide, apical flagellomere elongate with three stiff setae at apex. Mandible strongly sclerotized, evenly curved inward, (sickle-shaped), as long as head capsule. Labial palpus longer than mandible width, composed by at least three palpomeres. Surface of head capsule granulose, with papillae and covered with thin, minute, setae. Thorax nearly as wide as head. Pronotum strongly sclerotized, distinctly narrower than head capsule. Meso- and metathorax membranous. Legs relatively short and robust. Femora and tibiae similar-sized; tarsi relatively short, pretarsi with relatively short claws and arolia. Abdomen composed of at

least nine recognizable segments, gently tapering posteriorly. Abdominal segments 1–8 wider than long. Abdominal segment 9, elongate, longer than wide. Abdomen almost devoid of setae except stiff, hair-like setae at lateral and posterior margins of segment 8 and fine setae on apical 2/3rd of segment 9.

**Comments**. The presence of papillae on the head capsule, a relatively long antenna, toothless mandible, and labrum with prominent rostrum allow assignment of this specimen to Psychopsidae. This larva differs from all other known extant and fossil psychopsid larvae, including the coeval *Acanthopsychops*, owing to the trapezoidal rostrum with two erect paramedial setae, antennal flagellum composed of only four flagellomeres (>10 flagellomeres in other psychopsids), and a strongly sclerotized pronotum (membranous in other psychopsids).

Family Nymphidae Rambur, 1842
*Nymphavus progenitor* Badano, Engel & Wang gen. et sp. nov. (Fig. 3c, Supplementary Figs. 1h, 2e)
LSID:     urn:lsid:zoobank.org:act:22FAB2C1-339C-4612-B754-58B76D53B6D9;     urn:lsid:zoobank.org:act:97AD0BBE-2CE2-4522-9C6F-0FFBADD9B448.

**Type species**. *Nymphavus progenitor* Badano, Engel & Wang sp. nov.

**Etymology**. The generic name is a combination of Latin words: *Nympha* "young woman", commonly applied to split-footed lacewings, and *avus* "ancestor". It should be treated as masculine. The specific name, from Latin, means "ancestor".

**Type material**. Holotype: BA12015 excellent state of preservation, dorsal view only (Supplementary Fig. 1h). Probably 1st instar. Paratypes: AMNH JCZ-Bu33A (Fig. 3c, Supplementary Fig. 2e), specimen carrying a considerable amount of detritus, making difficult to appreciate the dorsal side of the body; NIGP164047, partly covered by detritus; NIGP164051 partly covered by detritus; NIGP164058, partly decomposed. Morphotype MI[23].

**Description (larva)**. Measurements (Supplementary Table 1). Head capsule quadrate, as long as wide. Temples well pronounced, giving head capsule a distinctly angular appearance. Stemmata sessile, placed on lateral surface of head capsule. Anterior margin of labrum with four median spiniform processes, each bearing an apical bristle-like sensillum. Gular sclerite subtriangular. Antenna relatively short, 1/2 as long as mandible, raised on a short tubercle. Scape cylindrical, distinctly differentiated in size from flagellomeres; pedicel as long as one flagellomere; flagellum filiform. Palpiger flattened. Labial palpus longer than mandibular width. Mandibles widely separated at base. Mandible slender, much longer than head capsule, curved inward, with one median tooth. Mandibular tooth much longer than mandible width. Numerous thin setae arranged along internal and external margins of mandible. Head capsule covered with thin bristle-like setae. Meso- and metathorax bearing one elongate, slender scolus-like setiferous process. Legs similar-sized, all podites articulated. Tibiae with a short ventral projection (1/8th of podite length) in proximity of articulation with tarsi. Tarsi short, as long as 1/3rd of length of tibiae; pretarsi bearing inconspicuous claws. Legs covered with thin, hair-like setae. Abdomen composed of nine visible segments. Abdominal segments 1–7 equipped with setiferous processes, of which dorsal series poorly preserved. Ventral series of setiferous processes scolus-like, progressively reduced in size toward abdominal apex. Abdominal segment 8 without processes. Abdominal segment 9 subconical, longer than wide.

**Comments**. *Nymphavus* closely resembles others of the nymphid subfamily Nymphinae, to which it belongs as evidenced by the

analysis. In overall morphology it is a typical nymphine but differs in the bristle-like sensillae covering the head capsule (instead of dolichasters) and the relatively elongate antenna, as long as 2/3rd of the mandible length (opposed to 1/3rd of the mandible length).

Stem-group Myrmeleontidae+Ascalaphidae
*Electrocaptivus xui* Badano, Engel & Wang gen. et sp. nov. (Fig. 3d, Supplementary Fig. 2b, c)
LSID: urn:lsid:zoobank.org:act:C124B69A-A5EB-40B3-9BB9-0495F7E25999; urn:lsid:zoobank.org:act:04E52BAA-4DB0-466F-8E51-A679AC979B72.

**Type species**. *Electrocaptivus xui* Badano, Engel & Wang sp. nov.
**Etymology**. The genus name is a combination of Latin words, meaning "amber prisoner" and is masculine. The species is named after Xianfeng Xu, who brought the specimen to the attention of the authors.
**Type material**. Holotype: BA17101 (Fig. 3d, Supplementary Fig. 2b, c), partly decomposed.
**Description (larva)**. Measurements (Supplementary Table 1). Head capsule square shaped with blunt margins. Posterior portion of head with angular temples. Stemmata sessile. Anterior margin of labrum slightly concave, with hair-like setae. Antenna extremely short, thin, slightly longer than mandibular width basally. Scape wider than flagellum. Palpiger flattened (remainder of palpus not preserved). Mandible relatively robust, as long as head capsule, evenly curved inward. Median internal margin of mandible with a multicusped lobe, with two tooth-like protrusions. External margin of mandible with thin bristles. Dorsal surface of head with sparse bristle-like setae. Lateral margins of head capsule with elongate hair-like setae, progressively longer toward posterior portion of head. Cervix tubular, strongly sclerotized, almost as long as head capsule. Surface of cervix with sparse bristle-like raised on wart-like papillae. Prothorax poorly preserved, not bearing traces of setiferous processes. Meso- and metathorax each with two lateral scolus-like setiferous processes with a hair-like seta at apex, much longer than process. Legs short and slender compared to body length, with all podites articulated. Pretarsal claws relatively small. Abdomen equipped with a dorsal and a ventral series of scolus-like setiferous processes, arranged along its lateral surfaces. Abdominal processes of dorsal series similar in shape, size, and chaetotaxy to those on thorax, bearing characteristic hair-like setae.
**Comments**. *Electrocaptivus* is characterized by an unusual combination of characters absent in any known larva of Neuroptera. The numerous autapomorphies of this species are: anterior margin of clypeolabrum with relatively long setae, lateral margin of the head capsule with relatively long setae, internal margin of the mandible with a multicusped lobe, thoracic and abdominal setiferous processes with extremely long setae, much longer than the processes. The elongate and sclerotized cervix resembles the condition observed in Crocinae (Nemopteridae), although *Electrocaptivus* notably differs from the species of this subfamily in having prominent setiferous processes.

*Burmitus tubulifer* Badano, Engel & Wang gen. et sp. nov. (Supplementary Fig. 2d)
LSID: urn:lsid:zoobank.org:act:B038405F-DFCF-46DD-BB03-7BA52CC1298E; urn:lsid:zoobank.org:act:2C3FF760-F6B3-461B-9F0C-C06CF3F0A9BD.

**Type species**. *Burmitus tubulifer* Badano, Engel & Wang sp. nov.
**Etymology**. *Burmitus* is a compound masculine name, composed by Burma (colonial name of Myanmar) and the suffix -*mitus*,

common to many ascalaphids. The epithet also resembles the word "burmite", the mineralogical name of Burmese amber. The specific name, i.e., tube-bearing, refers to the spiracles.
**Type material**. Holotype: BA12013 (Supplementary Fig. 2d), excellent state of preservation, partly hidden by detritus. 1st instar larva. Morphotype MIII[23].
**Description (larva)**. Measurements (Supplementary Table 1). Head capsule subrectangular, longer than wide. Posterior portion of head capsule cordate, temples smooth. Dorsal surface of head capsule with a shallow depression. Ocular region raised on a prominent tubercle, longer than wide, with a subcircular cross-section, bearing seven relatively large stemmata. Anterior margin of labrum concave with a fringe of setae. Antenna 2/3rd as long as mandible, borne on tubercle. Scape cylindrical, relatively robust; pedicel as long as one flagellomere, flagellum filiform, slightly thinner than scape diameter, composed of more than 10 short flagellomeres, apical flagellomere fusiform. Mandible longer than head capsule, relatively straight and gently curved inward, with three teeth. Distance between base of mandible and basal tooth equal to that between basal and apical teeth. Median tooth slightly closer to basal tooth than to apical tooth. Apical tooth larger that median and basal teeth. Mandible bare, except a few short setae on external margin. Neck region short, less than 1/3rd as long as head capsule. Cervix collar-like and membranous. Pronotum cuneiform, longer than wide. Mesothoracic spiracle tubular in shape, strongly sclerotized, exceptionally prominent, at 3 times longer than wide. Both meso- and metathorax with one elongate and slender scolus-like setiferous process, arranged on lateral surface. Legs long and slender (in comparison to body length), of similar size, with all podites articulated. Tibiae with a short spur in proximity of articulation with tarsi. Tarsi long and slender in comparison to tibiae. Pretarsal claws hook-shaped, divergent from each other. Legs covered with dolichasters. Abdomen composed of nine visible segments. Specimen shows a partly exposed segment 10. Lateral surface of abdomen with a dorsal series of long, scolus-like setiferous processes. Sternite 9 elongate, longer than wide.
**Comments**. Like other stem-group Myrmeleontidae+Ascalaphidae from Burmese amber, *Burmitus* has fully articulated metatibia and metatarsus but differs from the remainder of the group in having the antenna as long as 2/3rd of mandible length and only one pair of meso- and metathoracic setiferous processes. The latter character is shared with Nymphidae (including *Nymphavus*) and *Cladofer*. This species also differs from the coeval *Adelpholeon*, *Diodonthognathus*, and *Mesoptynx* in having three mandibular teeth instead of two. The elongate, tubular mesothoracic spiracle is an unusual diagnostic character of this species, allowing it to be differentiated from any other known myrmeleontiform larva.

*Diodontognathus papillatus* Badano, Engel & Wang gen. et sp. nov. (Fig. 3e)
LSID: urn:lsid:zoobank.org:act:CE4304C3-F243-490C-92A1-7D4210F7B55F; urn:lsid:zoobank.org:act:69EC5FE0-4103-454E-8397-F97608363DBF.

**Type species**. *Diodontognathus papillatus* Badano, Engel & Wang sp. nov.
**Etymology**. The generic name, from Greek, refers to the two-toothed jaws, and is masculine. The specific epithet is an adjective, meaning "bud-like", referring to the papillae on the head.
**Type material**. Holotype: BA12011 (Fig. 3e). Paratypes: BA12012; BA12014; BA12017; NIGP164042; NIGP164044; NIGP164046; NIGP164048; NIGP164049; NIGP164050; NIGP164059;

NIGP164060; AMNH JCZ-Bu29, only head and part of the thorax preserved, mostly decomposed. Morphotype MII[23].

**Description (larva).** Measurements (Supplementary Table 1). Head capsule subrectangular, longer than wide. Posterior portion of head cordate, with smooth temples. Dorsal surface of head capsule with shallow depression. Ocular region raised on prominent tubercle with a subcircular cross-section, with six dorsal and one ventral stemmata. Anterior margin of labrum concave, with cylindrical protuberances bearing setae. Ventral surface of head with a triangular gular sclerite and maxillary grooves arranged obliquely on anterior portion of head capsule. Antenna relatively short, less than 1/3rd as long as mandible, borne on a short tubercle. Scape cylindrical, relatively robust; flagellum filiform, noticeably thinner than scape, composed of more than 10 flagellomeres. Labial palpus 1/3rd as long as mandible, composed of three elongate palpomeres. Mandible slender, much longer than head capsule, curved inward, with two pointed teeth. Teeth longer than mandibular width, apical tooth slightly longer than basal tooth. Portion of mandible between base and basal tooth with at least eight thin setae, a similar number of interdental mandibular setae present between basal and apical teeth, a few short setae placed beyond apical tooth. External margin of mandible with a sparse fringe of setae reaching apical curve of mandible. Head capsule covered with sparse bristle-like setae raised on cylindrical protuberances. Ventral surface of head smooth. Cervix collar-like, membranous. Pronotum elliptical, longer than wide. Mesothorax with two long and slender scolus-like setiferous processes, of which anterior process 1/3rd as long as posterior process. Metathorax with only one thin, much longer than wide, scolus-like setiferous process preserved (but probably mirroring same arrangement of mesothorax in life). Legs relatively long and slender in comparison to body length, of similar size, with all podites articulated. Tibiae with a short spur in proximity of articulation with tarsi. Tarsi relatively long and slender, narrower than tibiae. Pretarsal claws relatively thin, divergent from each other. Legs covered with thin setae. Abdomen composed of nine visible segments. Abdomen with a dorsal series of relatively stout and rounded scolus-like setiferous processes, along its lateral surface. Abdominal sternite 9 longer than wide.

**Comments.** The main autapomorphy of *Diodontognathus* is the presence of cylindrical setae-bearing protuberances on the dorsal surface of the head capsule. This character sets this larva apart from all other Myrmeleontiformia, including the related *Adelpholeon* and *Mesoptynx*. The globose eye tubercles are also diagnostic of this species.

*Mesoptynx unguiculatus* Badano, Engel & Wang gen. et sp. nov. (Supplementary Fig. 2e)
LSID:     urn:lsid:zoobank.org:act:423F5638-55EE-4FAE-9515-7656A0E15DC3;     urn:lsid:zoobank.org:act:6AD0F8BD-21E1-4025-B162-A60F4B747D16.

**Type species.** *Mesoptynx unguiculatus* Badano, Engel & Wang sp. nov.
**Etymology.** The generic name is a compound Greek name, from "*meso-*", referring to the Mesozoic Era, and *ptynx*, eagle-owl, also used for some ascalaphids. The gender of the name is feminine. The species name *unguiculatus* means "having small claws" and refers to the enlarged pretarsal claws.
**Type material.** Holotype: NIGP164043 (Supplementary Fig. 2e), specimen slightly decomposed. Morphotype MV[23].
**Description (larva).** Measurements (Supplementary Table 1). Head capsule rectangular, longer than wide. Temples not prominent. Stemmata raised on a prominent, longer than wide, tubercle with a subcircular cross-section. Anterior margin of

labrum concave, with a fringe of bristle-like setae. Antenna 2/3rd as long as mandible, borne on tubercle. Scape cylindrical; pedicel as long as scape; flagellum filiform, noticeably thinner than previous articles, composed of more than 10 flagellomeres. Mandible slender, much longer than head capsule, curved inward, with two teeth longer than mandible width. Internal margin of mandible with several bristle-like setae. External margin of mandible with a sparse fringe of setae reaching apical curvature. Head capsule with bristle-like setae. Cervix collar-like, membranous. Pronotum elliptical, longer than wide. Mesothorax with two elongate scolus-like setiferous processes directed forward, of which anterior process is 1.5 times as long as posterior process. Metathorax with two, as long as wide, scolus-like setiferous processes. Legs long and robust in comparison to body, of similar size, with all podites articulated. Tibiae with a short spur in proximity of articulation with tarsi. Tarsi 2/3rd the length of tibiae. Pretarsal claws of all legs equally enlarged, strongly curved. Legs covered with thin setae. Abdomen with a dorsal series of relatively stout scolus-like setiferous processes along its lateral surface.

**Comments.** *Mesoptynx* differs from the remainder of the group Myrmeleontidae+Ascalaphidae in the enlarged pretarsal claws on all legs (only on the metathoracic leg in Myrmeleontidae). This species shares with *Diodontognathus*, with which it forms a clade, the bristle-like covering of the head capsule.

*Adelpholeon lithophorus* Badano & Engel gen. et sp. nov. (Fig. 3f, Supplementary Fig. 2f, g)
LSID:     urn:lsid:zoobank.org:act:3559ED6E-3733-4941-8048-5A86CE143F47;   urn:lsid:zoobank.org:act:55EC0B78-231D-45FB-894A-791902F73C66.

**Type species.** *Adelpholeon lithophorus* Badano & Engel sp. nov.
**Etymology.** The generic epithet is a compound, masculine name from Greek, meaning "lion brother", in reference to its phylogenetic position as sister to the antlion+owlfly clade. The species epithet means "stone bearer," as a reference to the camouflaging behaviour.
**Type material.** Holotype: AMNH JCZ-Bu31 (Fig. 3f, Supplementary Fig. 2f, g), state of preservation excellent but dorsal side of the body mostly hidden by the carried debris. Body proportion and chaetotaxy suggest that this specimen is a 2nd or 3rd instar.
**Description (larva).** Measurements (Supplementary Table 1). Head capsule subrectangular, longer than wide. Ocular region raised on a prominent, longer than wide tubercle with a subcircular cross-section, with six dorsal and one ventral stemmata. Clypeolabrum covering entire anterior margin of head. Anterior margin of labrum with a relatively deep median concavity with dolichasters. A thin projection present, covering base of mandible. Ventral surface of head with a relatively small, less than 1/5th of head length, triangular gular sclerite and with hypostomal bridge. Maxillary groove arranged obliquely in anterior portion of head capsule. Posterior portion of head capsule rounded. Antenna relatively short, at least 1/3rd of mandible length, borne on tubercle. Scape cylindrical, with an external fringe of setae; pedicel as wide as scape but much shorter; flagellum filiform, noticeably thinner than previous articles, composed of more than 10 flagellomeres. Palpiger flattened. Labial palpus much longer than mandibular width, composed of three elongate palpomeres. Mandible slender, much longer than head capsule, evenly curved inward and slightly narrower basally, bearing two pointed teeth. Teeth longer than mandibular width, apical tooth much longer than basal tooth. Portion of mandible between base and basal tooth almost bare. Space between basal and apical teeth with 3–4 setae, and 1–2 setae beyond apical tooth. External margin of mandible with a sparse fringe of setae reaching apical tooth. Head

capsule covered with erect dolichasters, shorter on ventral surface. Outlines of thorax almost completely hidden by detritus and sand particles. Legs long and slender in comparison to body length, not differentiated in size, all podites articulated. Tibiae with a short spur in proximity of articulation with tarsi. Tarsi 1/3rd as long as tibiae and much narrower. Pretarsal claws relatively thin, divergent from each other. Legs covered with thin setae. Abdomen composed of nine visible segments. First abdominal spiracle sessile, placed dorsally. Abdomen equipped with a dorsal and a ventral series of setiferous processes along its lateral surfaces. Abdominal processes of dorsal series scolus-like, but relatively short and stout. Processes of ventral series tubercle-like. Abdominal sternite 9 triangular in shape, much longer than wide, provided with small, but strongly sclerotized, rastra at apex. Each rastrum bearing four digging setae, thicker than those covering remainder of abdomen.

**Comments**. The morphology of this larva supports strong affinities with Myrmeleontidae and Ascalaphidae. Notably, *Adelpholeon* is characterized by the spiracle of the first abdominal segment placed dorsally, a condition also shared by some extant Ascalaphidae (e.g., *Haploglenius* Burmeister, *Ascalaphus* Fabricius). *Adelpholeon* is part of a clade, also including *Mesoptynx* and *Diodonthognathus*, characterized by the presence of two mandibular teeth. This condition is observed in exceedingly few extant genera of Myrmeleontidae, such as *Glenurus* Hagen. However, the members of this entirely fossil clade differ from the extant families owing to the fully articulated metatibia and metatarsus. The main autapomorphy of *Adelpholeon* is the presence of a prominent projection at the base of the mandible.

Stem-Myrmeleontidae Latreille, 1802
*Pristinofossor rictus* Badano & Engel gen. et sp. nov. (Fig. 3g, Supplementary Fig. 2h)
LSID: urn:lsid:zoobank.org:act:70F9F446-901D-45D9-BDFF-932AC3D1D5A2; urn:lsid:zoobank.org:act:6F03590D-FB17-44DE-8072-8BF531F29BB8.

**Type species**. *Pristinofossor rictus* Badano & Engel sp. nov.
**Etymology**. The generic epithet is a combination of the Latin words, *pristinus*, "ancient", and *fossor* "digger", in reference to its supposed digging behaviour, and is of masculine gender. The specific epithet, *rictus*, from Latin, means "gaping mouth" referring to spread open mandibles of the type specimen.
**Type material**. Holotype: AMNH JCZ-Bu304 (Fig. 3g, Supplementary Fig. 2h): relatively well-preserved specimen but mostly hidden by detritus and cracks. Body proportion and chaetotaxy suggest that this specimen is a 2nd or 3rd instar.
**Description (larva)**. Measurements (Supplementary Table 1). Head capsule subrectangular, posterior portion not visible. Anterior labral margin with a median concavity with a fringe of dolichasters. Mandible slender, longer than head, blackish, relatively straight and gently curved inward. Mandible with three slender teeth, longer than mandibular width. Median tooth closer to apical tooth than to basal tooth. Basal tooth half as long as others; median tooth slightly longer than apical tooth. At least five interdental mandibular setae between base of mandible and basal tooth, 3–4 setae between basal and median teeth, one seta between median and apical teeth. External margin of mandible with short setae reaching apical tooth. Labial palpus only partly visible, longer than mandible width. Lateral and ventral surfaces of head thickly covered with elongate dolichasters. Thorax mostly hidden. Pro- and mesothoracic legs relatively long and slender in comparison to body length, with articulated podites; pretarsal claws relatively thin, parallel to each other. Metathoracic leg more robust than other legs, metatibia and metatarsus not articulated,

partly fused. Meta-pretarsal claws of metathoracic leg enlarged and more robust than other pairs. Legs with thin setae. Abdomen with setiferous processes of dorsal series tubercle-like and thickly covered with long, pale setae. Abdominal sternite 9 triangular in shape, longer than wide, provided with small rastra at apex. Each rastrum bearing four elongate digging setae. Setae covering 9th sternite progressively stouter toward apex.

**Comments**. *Pristinofossor rictus* is well supported as belonging to Myrmeleontidae. In contrast to extant antlions *Pristinofossor* lacks robust digging setae on the ventral surface of sternite 9. *Pristinofossor* also differs from most extant antlions, with the exception of the members of the tribe Dendroleontini, in having abdominal segment 9 longer than wide and comparatively inconspicuous rastra. The shape of mandible, whereby the median tooth is longer than the apical tooth, serves to differentiate this fossil species from extant members of Dendroleontini. Nevertheless, the resemblance with Dendroleontini, although not indicative of a strict relationship, may hint to a similar life-style. Several genera of Dendroleontini (e.g., *Dendroleon* Brauer, *Cymothales* Gerstaecker, *Tricholeon* Esben-Petersen) are specialized inhabitants of cave-like microhabitats, including tree holes and rock overhangs.

**Phylogenetic analyses**. The phylogenetic trees obtained under implied weighting with a *k*-value of 9.219 (see Methods) were selected to illustrate the relationships among clades and as a basis to perform statistical correlation analyses and ancestral state reconstructions (Fig. 4). Under these conditions, the analyses yielded 4 trees with a tree length of 301 steps, a total fit of 95.368, a consistency index of 0.530, and a retention index of 0.851 (Supplementary Fig. 3).

Our results are congruent with previous morphological studies in corroborating the monophyly of Myrmeleontiformia (Fig. 4). Moth lacewings (Ithonidae) were reconstructed as their closest extant relatives, taking up a position as their sister group, also consistent with many prior studies[13]. Two Cretaceous taxa, *Macleodiella* and *Cladofer*, were assigned as stem-group Myrmeleontiformia forming a clade consistently recovered by all analyses as sister to the remaining myrmeleontiform clades. Monophyletic silky lacewings (Psychopsidae), including fossil taxa, i.e., the Cretaceous *Acanthopsychops* and *Aphthartopsychops* and the Eocene taxon *Propsychopsis* Krüger, were found sister to Myrmeleontoidea (i.e., Nymphidae (Nemopteridae (Ascalaphidae, Myrmeleontidae))). Split-footed lacewings (Nymphidae) emerged as monophyletic and composed of two subclades: Myiodactylinae and Nymphinae. Notably, the Cretaceous *Nymphavus* from Burmese amber clustered with the extant Australian genus *Nymphes* Leach and the extinct *Pronymphes* MacLeod from Baltic amber, supporting an affiliation with Nymphinae. Monophyletic spoon and thread lacewings (Nemopteridae) were resolved as the sister group to a clade comprising stem- and crown-group representatives of antlions (Myrmeleontidae) and owlflies (Ascalaphidae) (Fig. 4). This latter group consists of a grade of Mesozoic, extinct taxa (*Electrocaptivus*, *Burmitus*, and a clade comprising *Adelpholeon*, *Mesoptynx*, and *Diodontognathus*) from which the ancestor of antlion+owlfly arose (Fig. 4). Fossil owlflies from Cenozoic ambers, such as larvae of *Neadelphus* MacLeod (Baltic amber) and *Ululodes* Currie (Dominican amber), were reconstructed as members of modern ascalaphid lineages. Antlions were recovered as monophyletic in all analyses, forming a dichotomy with the closely related owlflies, a result contrasting with some recent studies that have suggested that ascalaphids are derived from among myrmeleontids[12]. *Pristinofossor* was reconstructed as sister to all the remaining antlions in all analyses, thus representing the oldest undisputed larva of this

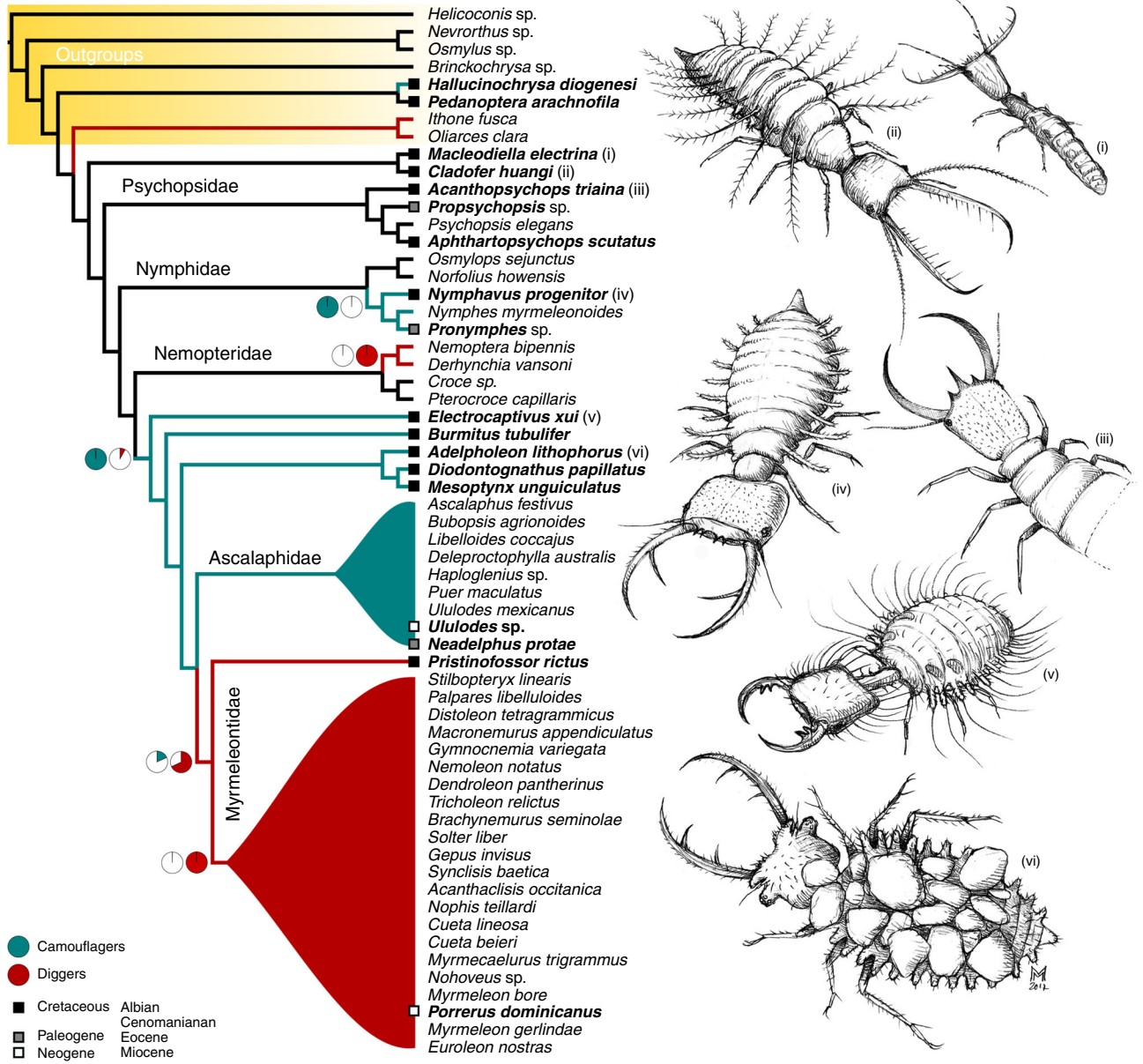

**Fig. 4** Phylogeny of Myrmeleontiformia, including fossils, based on larval morphology. The cladogram is a strict consensus tree of the four equally parsimonious trees obtained under implied weighting ($k$ = 9.219). Pie charts near nodes represent the proportional likelihood of digging and camouflaging behaviour across the Myrmeleontidae+Ascalaphidae clade calculated through ancestral state reconstruction. Terminal taxa marked in bold indicate amber-embedded specimens from mid-Cretaceous Burmese amber. Main extant families are collapsed. Drawings by Maurizio Mei

group (Fig. 4). While *Pristinofossor* agrees with modern antlions in overall morphology, the absence of specialized digging setae on the apical abdominal sternite suggests that it actually belongs to a myrmeleontid stem lineage. The larva of *Porrerus dominicanus* Poinar & Stange from Dominican amber was strongly supported as a member of the extant tribe Myrmeleontini.

**Correlation analyses.** We performed correlation analyses based on the output of the phylogenetic reconstructions to test whether: (i) camouflaging and digging behaviours are linked to selected morphological traits (Supplementary Fig. 4), and (ii) fossorial behaviour already characterized Mesozoic antlions. Therefore, we investigated whether key behavioural traits of extant species are significantly correlated to given morphological characteristics, thus aiming to infer the behaviour of fossil species (Table 1,

Supplementary Tables 2-6). We found a strong correlation between the presence of enlarged pretarsal claws and digging behaviour (Log Bayes Factor (LogBF) 9.820). Enlarged meta-thoracic legs also appear correlated with a fossorial life-style (LogBF 3.300). Unsurprisingly, the analysis supported a strong correlation between camouflaging behaviour and the presence of a dorsal series of elongate processes (i.e., scolus-like) on the abdomen (LogBF 18.880). We also recovered a correlation between camouflaging and the presence of a ventral series of elongate processes on the anteriormost abdominal segments (LogBF 8.530).

**Discussion**

Burmese amber gives a glimpse into a surprisingly diverse fauna of Mesozoic Myrmeleontiformia, revealing a morphological

**Table 1 Results of statistical correlation analyses between selected morphological and behavioural traits calculated from the mean values of the MCMC (LML) chains, repeated for 20 times, using Log Bayes Factor (LogBF)**

| Morphological character | Behavioural character | LML run mean | LML re-run mean | LogBF Mean | Significance |
|---|---|---|---|---|---|
| 74 Metathoracic leg | 114 Digging behaviour | −44.571 | −46.071 | 3.000 | Correlated |
| 78 Pretarsal claws | 114 Digging behaviour | −46.596 | −51.506 | 9.820 | Strongly correlated |
| 93 Abdominal segments 1–7, type of dorsal setiferous processes | 118 Camouflaging behaviour | −38.474 | −47.914 | 18.880 | Very strongly correlated |
| 94 First abdominal segments 1–2, type of ventral setiferous processes | 118 Camouflaging behaviour | −42.445 | −47.209 | 9.528 | Strongly correlated |
| 108 Rastra | 114 Digging behaviour | −54.253 | −46.450 | −15.607 | Weakly correlated |

Interpretation of statistical significance is also reported

diversity exceeding that of extant taxa. Our phylogenetic analysis convincingly assigns the diverse fossil larvae to both stem- and crown-groups, demonstrating that during the Early Cretaceous now extinct morphotypes lived alongside more modern forms. This remarkable fossil disparity allows for the refinement of character polarization and homology assessment in our matrix and, consequently, influenced the resolution of phylogenetic signal resulting from morphological evidence (Supplementary Note 1). Despite the long-recognized monophyly of Myrmeleontiformia, which has been corroborated in virtually every analysis, the relationships among the included families remain controversial, with a notable disagreement between molecular- and morphology-based phylogenies[11,12,31–33]. Phylogenomic studies[12] retrieved moth lacewings (Ithonidae) as nested within Myrmeleotiformia as part of a clade including Psychopsidae +(Nymphidae+Ithonidae), a surprising result which remains elusive using morphological traits alone as well as Sanger data[31,32]. The potential that ithonids are misplaced in phylogenomic studies remains to be explored. Our analysis did not recover such a relationship and instead strongly supports their position as sister to Myrmeleontiformia. Our results strongly agree with phylogenies based on Sanger data[14,32,33] resolving silky lacewings (Psychopsidae) as the earliest-diverging clade from the myrmeleontiform backbone among surviving families. Yet, in agreement with both mitochondrial phylogenomics[15] and Sanger data[14], split-footed lacewing (Nymphidae) are the following clade to branch off from the myrmeleontiform tree (Fig. 3), while thread- and spoon-winged lacewings (Nemopteridae) were found as sister to the strongly supported antlions+owlflies clade. Several cladistic analyses of morphological data, not employing extinct taxa, instead recover split-footed lacewings sister to antlions and owlflies[11,31,34,35]. Among the fossils, the silky lacewings *Acanthopsychops* and *Aphthartopsychops*, and the split-footed lacewing *Nymphavus*, were reconstructed as crown-group representatives, while the discovery of diverse antlion+owlfly stem-groups, such as *Electrocaptivus* and *Burmitus*, and the stem-group antlion *Pristinofossor*, characterized by unique combinations of morphological features, reveal a lively mosaic of "gains and losses" among these taxa. This pattern led to a reassessment of the phylogenetic informativeness of some characters, and resulted in a phylogeny more congruent with molecular lines of evidence. It is noteworthy that when Cretaceous fossils are removed from the analyses split-footed lacewings re-take their "traditional" position as the sister group to antlions and owlflies (Supplementary Fig. 5). This highlights the critical importance of extinct lineages for accurate phylogenetic assessment[36].

A major divergence between our results and all recent phylogenomic studies involved the resolution and the reciprocal relationships of the two largest and diverse groups of myrmeleontiforms: antlions and owlflies. Indeed, cladistic studies

invariably reconstructed these families as sister clades[11,37]. Studies based on Sanger data also recovered ascalaphids as monophyletic[14]. However, phylogenomic studies have retrieved antlions as paraphyletic with respect to owlflies, a seemingly unsurprising result considering the broadly overlapping *Bauplans* of these families[12,15].

The extraordinary diversity of mid-Cretaceous Myrmeleontiformia is also reflected in the numerous species represented as adults in this deposit, raising the interesting question of whether it would be possible to associate any of the larvae reported herein to those families, subfamilies, or genera already reported from Burmese amber. The extinct families Babinskaiidae and Araripeneuridae are both reported from Burmese amber[27,28,38]. Undescribed long-necked larvae from Burmese deposits were tentatively assigned to Araripeneuridae or to Babinskaiidae. Noteworthily, these "families" may be nothing more than stem-groups to existing myrmeleontiforms, and their utility and even validity remains to be critically tested[13]. Regardless, it is virtually impossible at present to definitively associate any of the larvae reported herein with species already characterized on the basis of adults. To do so would be entirely speculative, particularly in the absence of cladistic evidence for the proper phylogenetic placement of these putative families based on adults.

Larval myrmeleontiforms are easily recognizable from other neuropteran larvae because they have large, scythe-like jaws and a robust head capsule (Fig. 1d–f). However, they also show other striking features. For example, the larvae of split-footed lacewings, antlions, and owlflies are characterized by trap-like jaws armed with long, pointed teeth used to impale their prey. Moreover, on the lateral side of their rotund bodies, these larvae display an impressive complement of setae-bearing processes used for camouflaging or anchoring to hard surfaces (Fig. 1e, f). The larvae of thread lacewings (Nemopteridae: Crocinae) are unusual as well, due to their disproportionately long and thin neck-like thoracic anterior subsegment, which increases the mobility of the head and the strike range of the mandibles during ambush hunting[11,39,40] (Fig. 1d). The early stem-group myrmeleontiforms *Macleodiella* and *Cladofer* share long, multi-toothed mandibles (Fig. 3a, Supplementary Fig. 1a–e). *Cladofer* is also characterized by a series of tubular processes on the lateral side of the body (Supplementary Fig. 1d), which closely resembles those of Cretaceous chrysopoids, such as *Hallucinochrysa diogenesi* Pérez-de la Fuente et al. from Spanish amber[41,42] and several specimens from Burmese amber[23]. These findings support the view that these features are homoplastic among Neuroptera, and therefore their support of a clade comprising split-footed lacewings, antlions, and owlflies, as retrieved or hinted in previous studies[11,32,34], requires further testing. A character can perform well in one part of a phylogeny, but poorly in other parts of the tree, and so one cannot immediately use its presence elsewhere in

a phylogeny to deem it as not supporting a particular clade. Nonetheless, such homoplasy does stress the need for further inquiry, particularly into whether or not the trait is highly influenced by ecological forces, such as perhaps suggesting for this particular trait an adaptive role in this group of sit-and-wait predators.

An even more unusual combination of characters is displayed by *Electrocaptivus*, probably the most puzzling member of this extinct fauna. Its "crane-like" head–thorax articulation, combined with tooth-like projections on the medial margin of the mandible and the presence of elongate processes on the body, make this specimen a kind of terrific intermingling of myrmeleontiforms, exemplifying the morphological plasticity of this lineage (Fig. 3d). Moreover, the Cretaceous forest was also populated by diverse stem-group antlions+owlflies, whose members (i.e., *Burmitus, Mesoptynx, Diodonthognathus, Adelpholeon*) at a first glance closely resemble their extant relatives (Fig. 3e, f). However, closer inspection reveals that these larvae were characterized by fully articulated metathoracic tibiae and tarsi, while in crown-group taxa the metathoracic tibiae and tarsi are invariably and immovably fused. On the other hand, the remaining inclusions do not have significant differences from modern taxa, and such taxa would not seem out of place or time if observed alive today, reflecting the mosaic nature of the mid-Cretaceous fauna in which stem- and crown-group taxa once coexisted.

The rich fossil lacewing fauna from Burmese amber shows that myrmeleontiforms did not gain considerable morphological novelty during the subsequent 100 million years and their diversity seems to result from the different combinations of a limited set of character states in a complex trade-off game. This morphological stasis helped in reconstructing behaviours not otherwise preserved by a trace in the fossil record (e.g., many ephemeral biological or ethological traits do not leave a physical mark that is suitable for preservation). The inference of these behaviours allowed us to shed light on the ecological niche and life-style of extinct myrmeleontiforms. Our statistical correlation analysis strongly supported a correlation between the presence of strengthened metathoracic legs and fossorial behaviour, and again between the presence of enlarged pretarsal claws and fossorial behaviour, implying that these traits are indicators of fossorial behaviour in antlions (Table 1). *Pristinofossor*, the oldest known antlion larva, displays robust metathoracic legs and pretarsal claws (Fig. 3g), indicating that this species already had a fossorial way of life, like almost all living myrmeleontids[39,43] (Fig. 1h). While fossorial habits would be exceptionally difficult to preserve in the fossil record (perhaps only in the finest and most unusual settings among known paleosols), camouflaging behaviour often leaves a trace in amber, because the resin can trap larvae while still carrying debris on the body (Fig. 3c, e, f). Indeed, Burmese fossils show that debris-carrying characterized this lineage for at least 100 million years[23]. All of these camouflaging lacewings were equipped with elongate protuberances. The strong statistical correlation retrieved between the presence of these protuberances and camouflage (see Table 1) demonstrates that this trait is an indicator of such behaviour, even when the debris covering is not directly preserved in the amber piece together with the larva. Camouflaging and fossoriality appear widespread across Myrmeleontiformia, which is unsurprising in these ambush hunters and when considering that both behaviours allow the predatory larvae to hide from their unsuspecting prey, although in different ways (Fig. 1g, h). Debris-carrying is mostly performed by species living exposed on the upper soil layers, in foliage, or on hard surfaces (e.g., rocks, trunks) (Fig. 1g). Noteworthy, the fossil stem-myrmeleontiform *Macleodiella* does not show morphological characters correlated with fossoriality or camouflaging, while its sister taxon, *Cladofer*, displays extraordinary elongate body

protuberances, which are a clear clue of camouflaging behaviour (see above).

The results of the ancestral state reconstruction imply that camouflaging behaviour arose at least three times within the lineage, namely: in stem-group Myrmeleontiformia, in split-footed lacewings (Nymphinae), and in antlions+owlflies (Fig. 4). This behaviour also independently evolved in other unrelated lineages of neuropterans like the green lacewings (Chrysopidae) and their stem-groups[41,42,44,45]— incidentally known from the same deposit[22,23]. The scenario re-created through fossil evidence and the ancestral state reconstruction suggests that a camouflaging life-style might have characterized the ancestor of the myrmeleontid+ascalaphid clade, at the same time rejecting fossoriality with a high likelihood. Therefore, our analysis supports the notion that the fossorial antlions are secondarily derived from a camouflaging ancestor (Fig. 4). Interestingly, the fusion of tibia and tarsus in the metathoracic leg of the antlion+owlfly ancestor rendering the leg stiff and likely facilitating shovel-like movements appears to represent an exaptation to the fossorial life-style later exploited by the ancestor of antlions. Noteworthily, the fossorial specializations of ribbon lacewings (Nemopterinae) evolved independently, as also corroborated by major differences in morphology and behaviour[11]. Although this reconstruction is supported by both fossils and statistics, it should be noted that amber inclusions could be biased toward the preservation of arboreal camouflaging species. Most larvae from Burmese amber were probably tree-dwellers, squeezing among barks and crevices with their elongate bodies or hiding in plain sight under a cover of detritus. The fossorial life-style of antlions is certainly one of the factors leading to their success, allowing these insects to colonize and diversify in arid habitats where they survived the considerable changes in terrestrial environments during the Cretaceous[5,46], changes which presumably impacted, likely negatively, other co-existing myrmelontoid lineages. The winning specializations of antlions, today so intimately associated with deserts, first evolved in a wholly different environment, amid the shade of tropical coniferous forests in the Mesozoic.

## Methods

**Specimen origin**. All of the fossil materials described herein were extracted from the amber deposits located in Kachin Province, northern Myanmar, ca. 100 km west of the town of Myitkyina. The amber outcrops date back to the mid-Cretaceous, near the boundary between the Aptian and the Cenomanian[24,25]. Specimens are deposited in the following repositories: AMNH, American Museum of Natural History, New York, USA; BA, Lingpoge Amber Museum, Shanghai, China; NIGP, Nanjing Institute of Geology and Palaeontology, Nanjing, China. Inclusions were trimmed and polished using a water-fed lapidary wheel, for optimal views of each specimen.

**Imaging**. Specimens were photographed using a Nikon D3 DSLR camera with a Nikon SMZ1500 stereomicroscope, and processed using HeliconFocus software and Adobe Photoshop. Specimens were studied with both Nikon SMZ1500 and Olympus SZ12 stereomicroscopes.

**Phylogenetic analysis**. We implemented the data matrix developed as described in ref. [11], using the software Mesquite[47] including the fossil specimens from Mesozoic and Cenozoic ambers, such as Baltic amber (mid-Eocene) and Dominican amber (Early Miocene) (Supplementary Data 1). Thus, we analysed 60 taxa and 118 characters through parsimony analyses performed with TNT software (v1.6-beta)[48]. Characters were treated as unordered and analysed under implied and equal weights. We calculated the most fitting concavity $k$-value of the weighting function with the TNT script "setk.run"[49], obtaining $k = 9.219$. We ran the analysis using a "traditional search" algorithm, selecting the following settings: general RAM of 1.0 Gbyte, memory set to hold 1,000,000 trees, setting 1000 replicates with tree bisection reconnection (TBR[50]) branch swapping and saving 1000 trees per replicate. Zero-length branches were collapsed. Consistency index and retention index were calculated using the "Stats.run" script implemented in TNT. The parsimony analysis under equal weight produced 680 most parsimonious trees (301 steps) (Supplementary Fig. 6). Jackknife resampling was carried out under implied weights ($k$-value = 9.219) and equal weights with a traditional search producing 1000 replicates each of 1000 random taxa addition subreplicates

# ARTICLE

applying TBR branch swapping and saving 10 trees per replication; jackknife removal probability was set at the default value of 0.36. Bremer support values were calculated in TNT under equal weights from 10,000 trees up to 7 steps longer than the shortest trees obtained from a "traditional search", using the "trees from RAM" setting. Finally, we re-ran the analysis under implied weights after removing the fossil taxa to test their impact on the obtained topology (Supplementary Fig. 5).

**Tracking the correlation and the evolution of selected characters.** Character correlation and ancestral state reconstruction analyses were conducted for selected morphological and behavioural traits on the fittest trees obtained under implied weighting and k-value of 9.219, by choosing the number of character changes (steps) per branch as a measure of branch length. Character changes were optimized using unambiguous transformation algorithm. Trees with branch lengths proportional to the number of steps were obtained with TNT.

The correlation analyses were then carried out in BayesTraits V3, using the discrete module[51], with priors set to an exponential distribution with an average value of 10 and running each Markov chain Monte Carlo (MCMC) reversible-jump chain[52] for 1,000,000 iterations, sampling every 1000 iterations and with a burn-in of 10,000 iterations[51]. Discrete analyses compared dependent (correlated) and independent (not correlated) trait models calculated from the mean values of the MCMC chains, using LogBF[52,53]. Each correlation analysis was conducted 20 times and the resulting values were interpreted following ref. 54 (Supplementary Tables 26).

The ancestral state reconstructions of the digging behaviour (character 114) and camouflaging behaviour (character 118) were conducted with the "Multistate module" implemented in BayesTraits V3[55], with priors set to an exponential distribution with an average value of 10 and running the MCMC analyses for 1,000,000 iterations, sampling every 1000 iterations and setting a burn-in of 10,000 iterations[56]. Tags to identify all nodes were created using BayesTrees V1.3. Analyses were then visualized with TreeGraph 2[57].

**Nomenclatural acts.** This published work and the nomenclatural acts it contains have been registered in ZooBank, the proposed online registration system for the International Code of Zoological Nomenclature (ICZN). The ZooBank LSIDs (Life Science Identifiers) can be resolved and the associated information viewed through any standard web browser by appending the LSID to the prefix "http://zoobank.org/". The LSIDs for this publication are: urn:lsid:zoobank.org:pub:29BE2FC7-EC1F-4BAA-BDA9-C53BEEC2514A; urn:lsid:zoobank.org:act:B2B9C3F0-1341-47D2-AAFB-A8F57D16D0BA; urn:lsid:zoobank.org:act:3282500D-0015-4F2E-A0F8-FD6FFEDF89E9; urn:lsid:zoobank.org:act:DBB89A2E-7921-46C1-84CC-94F08E788C4D; urn:lsid:zoobank.org:act:06142ED1-80FE-4F5C-8A64-2ACFBF44459E; urn:lsid:zoobank.org:act:0323530A-94F2-464C-B6D9-6A572C512381; urn:lsid:zoobank.org:act:A9B11AFE-7281-4A3A-AD8F-77594316EEC2; urn:lsid:zoobank.org:act:D2567357-BDE5-4637-92DD-7A5CD16179E0; urn:lsid:zoobank.org:act:35CC68CC-2984-4950-8C9B-36A3C3397D5E; urn:lsid:zoobank.org:act:22FAB2C1-339C-4612-B754-58B76D53B6D9; urn:lsid:zoobank.org:act:97AD0BBE-2CE2-4522-9C6F-0FFBADD9B448; urn:lsid:zoobank.org:act:C124B69A-A5EB-40B3-9BB9-0495F7E25999; urn:lsid:zoobank.org:act:04E52BAA-4DB0-466F-8E51-A679AC979B72; urn:lsid:zoobank.org:act:B038405F-DFCF-46DD-BB03-7BA52CC1298E; urn:lsid:zoobank.org:act:2C3FF760-F6B3-461B-9F0C-C06CF3F0A9BD; urn:lsid:zoobank.org:act:CE4304C3-F243-490C-92A1-7D4210F7B55F; urn:lsid:zoobank.org:act:69EC5FE0-4103-454E-8397-F97608363DBF; urn:lsid:zoobank.org:act:423F5638-55EE-4FAE-9515-7656A0E15DC3; urn:lsid:zoobank.org:act:6AD0F8BD-21E1-4025-B162-A60F4B747D16; urn:lsid:zoobank.org:act:3559ED6E-3733-4941-8048-5A86CE143F47; urn:lsid:zoobank.org:act:55EC0B78-231D-45FB-894A-791902F73C66; urn:lsid:zoobank.org:act:70F9F446-901D-45D9-BDFF-932AC3D1D5A2; urn:lsid:zoobank.org:act:6F03590D-FB17-44DE-8072-8BF531F29BB8.

**Data availability.** The data supporting the findings of this study are detailed in the paper and its supplementary information files.

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

## Acknowledgements

Thanks are due to Cosmin-Ovidiu Manci (Tomesti, Romania) and Claudio Labriola (Portici, Italy) for sharing their photos. Special thanks to Maurizio Mei (Dipartimento di Biologia e Biotecnologie "Charles Darwin", Sapienza Università di Roma, Rome, Italy) for his lively and realistic reconstructions of fossil specimens and to Mara Tisato (Verona, Italy) for help in composing plates. Thanks also to Andrew Meade (University of Reading, UK) for his precious suggestions about his software, BayesTraits. The work of M.S.E. was supported by the US National Science Foundation grant DEB-1144162. B.W. was supported by the National Natural Science Foundation of China (41572010, 41622201, 41688103), and the Chinese Academy of Sciences (XDB26000000).

## Author contributions

D.B. and P.C. conceived and managed the project. M.S.E. and B.W. carried out taxon sampling and collection. D.B. and M.S.E. wrote the morphological descriptions. D.B. prepared the matrix. D.B., A.B. and P.C. analysed the data. D.B., M.S.E. and P.C. wrote the paper. All authors reviewed the manuscript.

## Additional information

**Competing interests:** The authors declare no competing interests.

