## [Peer Review File · Nature Communications]

Reviewers' comments:

Reviewer #1 (Remarks to the Author):

This is an important paper, reporting many exquisite Burmese myrmeleontiform larval ambers. It is evident that the lineages of Myrmeleontiformia are remarkably diverse in the Cretaceous Burmese, and it could also be deduced from the six morphotypes of Myrmeleontiformia in ref. 16. I approve of the authors' perspective to infer the evolutionary history under the framework of phylogeny. Some comments were made referring to the phylogenetic reconstruction integrated both extant and extinct insects and statistical correlation analysis on inference of larval ethology in this paper.

Phylogenetic analysis: This is a significant attempt to infer the systematic placements of the extinct lineages under the framework of phylogenetic analysis. The core of this method is depended on the accuracy of phylogenetic results. Although the authors adapted the implied weighting and seemingly retrieved the best supported MP trees (see Figure S1), it is also clear that the monophyly and inner-relationships within 'Total group Myrmeleontidae + Ascalaphidae' were not well reconstructed with the firm supports (see Supplementary Data 1). I notice that many missing data within the extinct lineages occurs in the matrix, and in my experience this condition could possibly lead to the inaccurate/artificial placements for the referring taxa. Maybe the authors could reduce the capacity of dataset to increase the contributions of fossil taxa during the analysis, or give more tests to improve the reliability of phylogeny.

Correlation analyses: I am very interesting in this section. The authors provided a clever and significant method to infer the behaviors of fossil insects. In the paper, the authors directly conducted the correlation analyses only using the sampled taxa of phylogenetic analysis, and I think this sample size is a little no enough to fully interpret this question. It is also likely that these morphological characteristics are just meaningful to the taxonomic traits instead of the indicator of behavior. If the correlation analysis were based on the extant insects with digging and camouflage behaviors, I believe it would be more persuasive than present.

Other comments:

1) Suggest the authors readjust the figures of plate 2 (Fig. 2). The citations seem very disordered, and it is a bit inconvenient for the readers.

2) In Supplementary Data 1, I notice some nodal synapomorphies are not consistent with Fig. S1, e.g. the clade of Crown-group Myrmeleontidae + Ascalaphidae, suggest the authors could recheck on this section.

Reviewer #2 (Remarks to the Author):

The current manuscript is an ambitious endeavour to incorporate larval Neuroptera into an analysis of phylogenetic relationships and evolutionary transitions. Larvae are notoriously rare in the fossil record. Even more, larvae are notoriously seen as "secondary" particularly in the taxonomic process, and the majority of species is described based on adults. In Myrmeleontidae, larvae have always played an important role in understanding the evolution and phylogeny of the group, and there is sufficient data available as a framework for this analysis. Describing a number of species based exclusively on fossil larvae is unusual, and one of the future challenges is that there will never be positive evidence to associate them with adults. Anyway, I think that the paper and its innovative combination of larval taxonomy and evolutionary interpretation is important and relevant and should be published as it is. If the entire material (descriptions and images) should be placed in the main text or in online supplementary material has to be decided by the editors. So in total, it is a very good manuscript!

Reviewer #3 (Remarks to the Author):

This manuscript represents two interesting and unique vantage points into evolutionary biology, both from a broad methodological and taxon-specific perspective. The first perspective relates to the utility of ontogenetically distinct morphological characters as well as fossil terminals in general in phylogenetic reconstruction while the second relates to the particular biology and evolutionary history of myrmeleontiforms. Badano and colleagues present a rich fossil assemblage and use these specimens to construct a convincing reconstruction of early antlion/lacewing life history. Overall, the methodology is sound and relatively conservative, however I have some fairly minor concerns related to the presentation of this work as well as the phylogenetic methodology. I recommend this paper for publication, following a response to the following:

What were the results of the equal weights tree and why is this topology not presented?

Line 78: regarding diversity a result of the age of the lineage, is there any evidence for this? For example, one could imagine other very old lineages such as horseshoe crabs with little diversity and younger diversification events leading to high diversity such as in teleost fish.

Line 113: are there any species with larvae that are not campodeiform in this group? If not, then it is not a particularly helpful diagnostic feature. Also, "Diagnosis" should indicate that it is for the larval form.

Line 126: No holotype included for *Cladofer huangi*. Type must be designated.

Line 241: What was the rationale for presenting the $k=13$ tree? Based on the stats provided for each k value, including the number of trees, total steps, and total fit, I do not see how the value of 13 stands out. Was it selected based on CI and RI? One thing to consider, there is a script called *setk* that is capable of searching for optimal K values given your own dataset. While it is nice to explore multiple k -settings, you should be explicit in selecting your preferred tree (*setk* described here: "Santos BF, Payne A, Pickett KM, Carpenter JM. (2014) Phylogeny and historical biogeography of the paper wasp genus *Polistes* (Hymenoptera: Vespidae): implications for the overwintering hypothesis of social evolution. *Cladistics* 31(5): 535-549."). If you are recovering significantly different trees under *setk*, I would suggest including a sensitivity matrix diagram for each node that is sensitive to different k values.

Line 242: You state that the topology is broadly consistent with trees recovered from phylogenomic work - in what ways is your topology in alignment and disagreement? You mention a few differences in the discussion, are these the only differences? It is difficult to assess how similar or different the topology is, given this broad statement.

Line 265: Avoid using the phrase "derived" as it is relative and somewhat arbitrary (ie are Diptera or Lepidoptera the more derived insects? Are Hymenoptera more derived than lice?). Could instead say that the genera with Cenozoic fossils, *Neadelphus* and *Ululodes*, were recovered within the modern subfamily *Ascalaphidae*.

Line 268: the position of *Pristinofossor* as sister to *Myrmeleontidae* is interesting and deserves commentary re: whether or not *Pristinofossor* actually falls into the family as described or represents a stem lineage of the group. Does *Pristinofossor* possess all synapomorphies for *Myrmeleontidae*? If not, should the synapomorphies of *Myrmeleontidae* be revised or should *Pristinofossor* be described as a stem lineage? You list the apomorphies for the family + *Pristinofossor* in the supplemental data, however it is not clear if these matrix-based characters are utilized to define clades outside of this analysis.

Line 272: The supplemental excel file for the correlation analyses is vague: I assume the first and second tab values (for example, 53-116) represent the two characters that are being examined, even though they are not labeled, however what is "ASR114"? Is this the ancestral state reconstruction and, if so, is this just the coding that is present in the morphological matrix itself?

Why relist those characters here?

Line 290: improved resolution compared to what? The resolution obtained prior to including these taxa in the matrix? Is this tree more resolved than prior analyses that did not include these new fossil species, and, if so, did the previous analyses implement the same implied weights assumptions? See comment on line 312 below.

Line 297: Before you state that your topology and that of phylogenomic analyses are largely congruent - this is an example of how it would be helpful to be explicit in what nodes are supported under both datasets and what nodes are different.

Line 312: There is no indication elsewhere in the paper that phylogenetic relationships were assessed with and without specific taxa. Is this with respect to another publication? If so, was implied weighting used in the same way? If this separate analysis was performed for this paper, the results (tree outputs) need to be included.

Line 358: Would say you don't need "putative" here, there is a lot of plasticity!

Line 417: this is a compelling sentence but should be reworded for clarity.

Line 437: should be "we ran"

--More Minor items and suggestions:

Throughout: to improve the general interest in this paper, I would recommend removing some of the higher taxonomic framing, when not using specific taxon names. For example, line 61: "not only have larvae been vital to resolving relationships among families, subfamilies, and genera, but" would be more succinct and equally informative were it "not only have larvae been vital to resolving relationships among neuropteran lineages, but"

It would be nice to include a timescale on the phylogeny presented. I realize that the divergence dates are not calibrated, however you could place the fossil terminals along a timescale and just note that the node ages are not based on any analyses.

Your description and the intuitive nature of the correlated traits you analyzed would be more effective with a generalized anatomical figure highlighting the characters that you assessed.

Line 35: repetition could be cut with respect to the terms "diverse" and "considerable diversification" in the same sentence.

Line 46: "The immature stages of insects embody a wealth of biological information, particularly so for holometabolous insects whereby the larval stages can often lead dramatically different lives from those of their corresponding adults." The beginning line is an example of not linking this paper with previous research. There is a tremendous amount of literature devoted to semaphoronts, phylogenetic signal and paleontological value of ontogenetically distinct morphotypes, as well as niche partitioning among holometabolous developmental stages. It would be nice to mention this elements in this paper, and potentially would cast a larger net with respect to readership in the paleontological and systematics community.

Line 167: recommend using more descriptive terminology than "squarish" - square-shaped with rounded margins, etc.

Line 289: should be "remarkable fossil disparity allows FOR the refinement..."

Recommend fewer - or no - instances of "in fact" in introductory text.

Responses to Reviewers' comments:

Reviewer #1 (Remarks to the Author):

This is an important paper, reporting many exquisite Burmese myrmeleontiform larval ambers. It is evident that the lineages of Myrmeleontiformia are remarkably diverse in the Cretaceous Burmese, and it could also deduced from the six morphotypes of Myrmeleontiformia in ref. 16. I approve of the authors' perspective to infer the evolutionary history under the framework of phylogeny. Some comments were made referring to the phylogenetic reconstruction integrated both extant and extinct insects and statistical correlation analysis on inference of larval ethology in this paper.

- We thank the reviewer for his/her careful attention and his/her precious insights on the manuscript.

Phylogenetic analysis: This is a significant attempt to infer the systematic placements of the extinct lineages under the framework of phylogenetic analysis. The core of this method is depended on the accuracy of phylogenetic results. Although the authors adapted the implied weighting and seemingly retrieved the best supported MP trees (see Figure S1), it is also clear that the monophyly and inner-relationships within 'Total group Myrmeleontidae + Ascalaphidae' were not well reconstructed with the firm supports (see Supplementary Data 1). I notice that many missing data within the extinct lineages occurs in the matrix, and in my experience this condition could possibly lead to the inaccurate/artificial placements for the referring taxa. Maybe the authors could reduce the capacity of dataset to increase the contributions of fossil taxa during the analysis, or give more tests to improve the reliability of phylogeny.

- Thanks, this is a good point. We re-ran analyses after having inactivated all characters with lots of missing data for extinct taxa following the reviewer suggestion. Interestingly, this did not alter the results if not for distal branches in particular within Ascalaphid and Myrmeleontid clades. Discussing such distal differences was out of our target, we then decided to stay with the original dataset.

Correlation analyses: I am very interesting in this section. The authors provided a clever and significant method to infer the behaviors of fossil insects. In the paper, the authors directly conducted the correlation analyses only using the sampled taxa of phylogenetic analysis, and I think this sample size is a little no enough to fully interpret this question. It is also likely that these morphological characteristics are just meaningful to the taxonomic traits instead of the indicator of behavior. If the correlation analysis were based on the extant insects with digging and camouflage behaviors, I believe it would be more persuasive than present.

- We thank the reviewer for his/her appreciation. The sample size that we analysed actually reflect the present state of knowledge of the larvae of Myrmeleontiformia, because all the known extant morphotypes, combination of characters and performed behaviours are represented. Moreover, all families, subfamilies and main tribes are included in the dataset. Since our aim is to reconstruct the phylogenetic relationships of fossil larvae and to infer their behavior, we think that our taxon sampling is representative enough to answer these questions. Moreover, the analysed morphological characters clearly transcend the simple taxonomic value. For example, elongate abdominal processes occur in different families and they are homoplasious. Similarly, enlarged pretarsal claws are present in unrelated families (i.e. Myrmeleontidae and Ithonidae) but they are positively correlated with a fossorial life-style. The correlation analyses were actually based on extant insects because these behavioural traits are usually not preserved in the fossil records. However, the camouflaging behavior is a notable exception, because the larvae remained untangled in resin while still carrying their camouflage on the dorsum (just look at the figures). We calculated the correlation between selected morphological traits and the behavior in extant lacewing larvae and then we inferred the presence of this behavior in extinct species based on the presence of the same morphological character. Then, we interpreted the results based on the statistical significance. The results are highly convincing based on the present knowledge of the larvae of Neuroptera and discussed and treated only if statistically relevant.

Other comments:

1) Suggest the authors readjust the figures of plate 2 (Fig. 2). The citations seem very disordered, and it is a bit inconvenient for the readers.

- We understand that plate 2 is confusing due to the very long figure caption which contains too much data (e.g. taxonomic placement, type status, specimen code...). We modified the figure caption following this comments.

2) In Supplementary Data 1, I notice some nodal synapomorphies are not consistent with Fig. S1, e.g. the clade of Crown-group Myrmeleontidae + Ascalaphidae, suggest the authors could recheck on this section.

- We thank the reviewer for noticing this error in the text of the supplementary files. We checked everything and corrected accordingly.

Reviewer #2 (Remarks to the Author):

The current manuscript is an ambitious endeavour to incorporate larval Neuroptera into an analysis of phylogenetic relationships and evolutionary transitions. Larvae are notoriously rare in the fossil record. Even more, larvae are notoriously seen as "secondary" particularly in the taxonomic process, and the majority of species is described based on adults. In Myrmeleontidae, larvae have always played an important role in understanding the evolution and phylogeny of the group, and there is sufficient data available as a framework for this analysis. Describing a number of species based exclusively on fossil larvae is unusual, and one of the future challenges is that there will never be positive evidence to associate them with adults. Anyway, I think that the paper and its innovative combination of larval taxonomy and evolutionary interpretation is important and relevant and should be published as it is. If the entire material (descriptions and images) should be placed in the main text or in online supplementary material has to be decided by the editors. So in total, it is a very good manuscript!

- We deeply thank the reviewer for his/her comments, greatly improving the manuscript. In endopterygote insects a correct match larva-adult is possible only through direct rearing or barcoding, thus it is virtually impossible (at present) to associate the larva to the respective adult in fossils. We thought a lot about this problem. So, we opted to consider each larval morphotype as a distinct taxonomic unit worth of a formal binomial naming and description since it was encased in a sound phylogenetic framework. Each coined name is informative because it unequivocally refers to a species placed in a cladistic context. The use of other solutions, such as alphanumeric codes is not feasible in a such diverse fauna, and we would fall in a pre-Linnean chaos... very inconvenient to the readers. Taxonomic names are important if informative, and this is also the case of these fossil specimens. Anyway, describing species on larvae is not unusual at all, especially in palaeontology. Here are some notable examples:

Chen J, Wang B, Engel MS, Wappler T, Jarzembowski EA, Zhang H, Wang X, Zheng X, Rust J. 2014. Extreme adaptations for aquatic ectoparasitism in a Jurassic fly larva. *eLife* 3:e02844

Huali C, Muona J, Hanyong P, Li X, Chen W, Teräväinen M, Dong R, Qiang Y, Xingliao Z, Songhai J. (2016) Chinese Cretaceous larva exposes a southern Californian living fossil (Insecta, Coleoptera, Eucnemidae). *Cladistics* 32:2, 211-214.

MacLeod, E. G. The Neuroptera of the Baltic Amber. I. Ascalaphidae, Nymphidae, and Psychopsidae. *Psyche*, 77, 147–180 (1970).

Pérez-de la Fuente, R. et al. Early evolution and ecology of camouflage in insects. *Proc. Natl. Acad. Sci. U.S.A.* 109, 21414–21419 (2012).

Wang B., Zhang. H. (2010), Earliest Evidence of Fishflies (Megaloptera: Corydalidae): An Exquisitely Preserved Larva from the Middle Jurassic of China. *Journal of Paleontology*, 84(4):774-780.

Reviewer #3 (Remarks to the Author):

This manuscript represents two interesting and unique vantage points into evolutionary biology, both from a broad methodological and taxon-specific perspective. The first perspective relates to the utility of ontogenetically distinct morphological characters as well as fossil terminals in general in phylogenetic reconstruction while the second relates to the particular biology and evolutionary history of myrmeleontiforms. Badano and colleagues present a rich fossil assemblage and use these specimens to construct a convincing reconstruction of early antlion/lacewing life history. Overall, the methodology is sound and relatively conservative, however I have some fairly minor concerns related to the presentation of this work as well as the phylogenetic methodology. I recommend this paper for publication, following a response to the following:

- We really thank the reviewer for his/her valuable comments which, we believe, tremendously improved the manuscript.

What were the results of the equal weights tree and why is this topology not presented?

- The 4 trees obtained under implied weights ($k=9.919$, see following comment) are characterized by the same tree length of those resulting from the analysis under equal weights, so they are included among the 680 trees obtained enforcing the latter conditions. In this way, we selected the 4 most fitting tree among the 680 shorter trees. However, following the reviewer's suggestion, we included the strict consensus tree obtained under equal weight as supplementary information.

Line 78: regarding diversity a result of the age of the lineage, is there any evidence for this? For example, one could imagine other very old lineages such as horseshoe crabs with little diversity and younger diversification events leading to high diversity such as in teleost fish.

- We wanted to emphasize the high morphological disparity of the group with respect to the relatively small number of extant species. However, we agree with the reviewer and it is highly problematic to attribute these differences exclusively to the ancient origin of the group. We accepted his/her suggestion and removed the ambiguous sentence.

Line 113: are there any species with larvae that are not campodeiform in this group? If not, then it is not a particularly helpful diagnostic feature. Also, "Diagnosis" should indicate that it is for the larval form.

- All the larvae of Neuroptera are basically campodeiform (the only known exception are Ithonidae which are melolonthoid), so we agree that this feature is not particularly informative in the diagnosis and we removed it, as suggested. We also accepted to specify that the diagnosis refers to the larva.

Line 126: No holotype included for Cladofer huangi. Type must be designated.

- We absolutely agree. When we submitted the first draft, the type was still in progress to be formally deposited in a public collection and still in need to receive an univocal specimen code. We have now obtained these fundamental data (i.e. type code and collection depository) so we included them in the text.

Line 241: What was the rationale for presenting the $k=13$ tree? Based on the stats provided for each k value, including the number of trees, total steps, and total fit, I do not see how the value of 13 stands out. Was it selected based on CI and RI? One thing to consider, there is a script called setk that is capable of searching for optimal K values given your own dataset. While it is nice to explore multiple k -settings, you should be explicit in

selecting your preferred tree (setk described here: "Santos BF, Payne A, Pickett KM, Carpenter JM. (2014) Phylogeny and historical biogeography of the paper wasp genus Polistes (Hymenoptera: Vespidae): implications for the overwintering hypothesis of social evolution. Cladistics 31(5): 535-549."). If you are recovering significantly different trees under setk, I would suggest including a sensitivity matrix diagram for each node that is sensitive to different k values.

- This is a good point and we really thank the reviewer for raising it and for his/her very constructive suggestions. Our rationale to select a k=13 was due not only because it yielded a topology broadly consistent with phylogenomic studies but also because under similar k-settings, implied weights analyses outperforms equal weights (see Goloboff PA, Torres A, Salvador Arias J (2017) Weighted parsimony outperforms other methods of phylogenetic inference under model appropriate for morphology. Cladistics doi:10.1111/cla.12205). Following the reviewer's suggestions, we calculated the more appropriate k value using the suggested TNT script "setk.run", which yielded a k=9.2188. The trees obtained under "setk" are consistent and perfectly comparable with the previous ones, confirming the same topology. In any case, we positively accepted this suggestion and adopted it throughout the manuscript, because this TNT-script allows to apply a less arbitrary and much more statistically sound selection of the k setting. Of course, we re-ran all the analyses under the new k-value. All relationships and results were again confirmed.

Line 242: You state that the topology is broadly consistent with trees recovered from phylogenomic work - in what ways is your topology in alignment and disagreement? You mention a few differences in the discussion, are these the only differences? It is difficult to assess how similar or different the topology is, given this broad statement.

We have briefly mentioned all relevant differences between our results and those obtained with genomic data in the Discussion section, by adding a couple of sentences. In addition, we deleted the sentence mentioned by the reviewer "*the topology is broadly consistent with trees recovered from phylogenomic work*" because we chose to select the k-value of the weighting function using the TNT script setk.run, see comment above.

Line 265: Avoid using the phrase "derived" as it is relative and somewhat arbitrary (ie are Diptera or Lepidoptera the more derived insects? Are Hymenoptera more derived than lice?). Could instead say that the genera with Cenozoic fossils, Neadelephus and Ululodes, were recovered within the modern subfamily Ascalaphidae.

- We completely agree with the reviewers about the misleading use of the term "derived". We originally meant to describe the phylogenetic position of *Neadelephus* and *Ululodes* with respect to the phylogenetic tree and not to express a qualitative view about their relationships. We have removed this ambiguous word from the sentence.

Line 268: the position of Pristinofossor as sister to Myrmeleontidae is interesting and deserves commentary re: whether or not Pristinofossor actually falls into the family as described or represents a stem lineage of the group. Does Pristinofossor possess all synapomorphies for Myrmeleontidae? If not, should the synapomorphies of Myrmeleontidae be revised or should Pristinofossor be described as a stem lineage? You list the apomorphies for the family + Pristinofossor in the supplemental data, however it is not clear if these matrix-based characters are utilized to define clades outside of this analysis.

- We understand that there was some ambiguity in the text regarding the affinities of *Pristinofossor*, thus we implemented the manuscript following the reviewer's comments. Anyway, *Pristinofossor* share several synapomorphies with extant Myrmeleontidae but, lacks one significant, non-homoplasious, apomorphy (i.e. presence of specialized digging setae on abdominal sternite 9) and two homoplasious apomorphies of the crown-group. Taking into account the results of the cladistics analysis, *Pristinofossor* lies outside the crown group proper and it perfectly fits with the definition of stem-lineage. The results of the analysis were indeed used to delimit clades (extant taxa only) in a previous publication (see Badano, D., Aspöck, U., Aspöck,

H. & Cerretti, P. Phylogeny of Myrmeleontiformia based on larval morphology (Neuropterida: Neuroptera). Syst. Entomol. 42, 94–117 (2017)).

Line 272: The supplemental excel file for the correlation analyses is vague: I assume the first and second tab values (for example, 53-116) represent the two characters that are being examined, even though they are not labeled, however what is "ASR114"? Is this the ancestral state reconstruction and, if so, is this just the coding that is present in the morphological matrix itself? Why relist those characters here?

- The supplemental excel file was used to ran the statistical analyses with Bayes Traits, and then it was implemented with the outputs. However, the reviewer is right, this file is not particularly informative or helpful to the reader. We improved it, converting it into a table and adding labels for each character. ASR114 indeed refers to the coding of ancestral state reconstruction. We removed these lines following the reviewer's suggestion.

Line 290: improved resolution compared to what? The resolution obtained prior to including these taxa in the matrix? Is this tree more resolved than prior analyses that did not include these new fossil species, and, if so, did the previous analyses implement the same implied weights assumptions? See comment on line 312 below.

- This line refers to the differences in topology between the results of the present analyses and previous morphology based studies not including fossil data (ref. 11, 31, 34, 35, 39), whose results are quite divergent from DNA-based phylogenies regarding the reciprocal affinities of some families (e.g. Nemopteridae). These differences are thoroughly discussed in the text. We agree that the term "improved" is highly subjective and particularly misleading under this respect, since all phylogenetic analyses are hypotheses. We replaced it with a more neutral word. Finally, we added a supplementary figure (Fig. S3) showing the topology resulting after the fossil taxa are removed from the analysis. This particular cladogram is consistent with the aforementioned morphology-based studies. See also comment on line 312.

Line 297: Before you state that your topology and that of phylogenomic analyses are largely congruent - this is an example of how it would be helpful to be explicit in what nodes are supported under both datasets and what nodes are different.

- We thank the reviewer for highlighting this particular point. We modified the discussion section about the retrieved relationships to better summarize the similarities or differences with previous studies.

Line 312: There is no indication elsewhere in the paper that phylogenetic relationships were assessed with and without specific taxa. Is this with respect to another publication? If so, was implied weighting used in the same way? If this separate analysis was performed for this paper, the results (tree outputs) need to be included.

- Indeed, we tested the topology resulting from the removal of fossil taxa from the matrix. Accepting the proposal of the reviewer, we included the cladogram resulting from the analysis of the matrix pruned of fossils under implied weights (same k-values of the main analysis) as supplementary figure S3.

Line 358: Would say you don't need "putative" here, there is a lot of plasticity!

- Corrected.

Line 417: this is a compelling sentence but should be reworded for clarity.

- Noted and corrected accordingly.

Line 437: should be "we ran"

- Corrected.

--More Minor items and suggestions:

Throughout: to improve the general interest in this paper, I would recommend removing some of the higher taxonomic framing, when not using specific taxon names. For example, line 61: "not only have larvae been vital to resolving relationships among families, subfamilies, and genera, but" would be more succinct and equally informative were it "not only have larvae been vital to resolving relationships among neuropteran lineages, but"

- We agree with the reviewer and we limited the use of higher taxonomic ranks throughout the text.

It would be nice to include a timescale on the phylogeny presented. I realize that the divergence dates are not calibrated, however you could place the fossil terminals along a timescale and just note that the node ages are not based on any analyses.

- Since we did not calibrate the divergence times of the lineages of Myrmeleontiformia (see ref. 12 and 15 for a thorough reconstruction of this particular subject), we do not think that the inclusion of such a data would be particularly useful, since it would be a mostly aesthetic choice. Moreover, the age of the amber deposits where the investigated fossil were found (i.e. Burmese, Baltic and Dominican ambers) are well known and widely discussed in literature.

Your description and the intuitive nature of the correlated traits you analyzed would be more effective with a generalized anatomical figure highlighting the characters that you assessed.

- We added a figure as supplementary information (Supplementary Figure S5) to show the morphological traits used in the correlation analyses, as suggested by the reviewer.

Line 35: repetition could be cut with respect to the terms "diverse" and "considerable diversification" in the same sentence.

- Corrected, by using a synonym.

Line 46: "The immature stages of insects embody a wealth of biological information, particularly so for holometabolous insects whereby the larval stages can often lead dramatically different lives from those of their corresponding adults." The beginning line is an example of not linking this paper with previous research. There is a tremendous amount of literature devoted to semaphoronts, phylogenetic signal and paleontological value of ontogenetically distinct morphotypes, as well as niche partitioning among holometabolous developmental stages. It would be nice to mention this elements in this paper, and potentially would cast a larger net with respect to readership in the paleontological and systematics community.

- We agree with reviewer: the introductive sentences were relatively devoid of citations on such an important topic. We improved this section by adding references to some relevant studies on this subject, such as:

Line 167: recommend using more descriptive terminology than "squarish" - square-shaped with rounded margins, etc.

- Corrected.

Line 289: should be "remarkable fossil disparity allows FOR the refinement..."

- Corrected.

Recommend fewer - or no - instances of "in fact" in introductory text.

- Corrected.

REVIEWERS' COMMENTS:

Reviewer #1 (Remarks to the Author):

I accept the authors' modifications. This is an important contribution to understand the earlier evolution of neuropteran larva. I recommend publication of the paper as it stands.

Reviewer #3 (Remarks to the Author):

Following up on my initial review, the authors have sufficiently responded to my comments and I look forward to seeing the manuscript published. A very nice set of paleontological descriptions coupled with phylogenetics. Below are my minor comments, however, I do not think that these should warrant a new round of reviews on their own, unless the phylogenetic questions inspire doubt in the authors claims.

Line 36: "both stem- and crown-groups to extant families" should be "both stem- and crown-groups with respect to extant families" – not possible to be a crown group *to* modern families.

Line 110: while I did recommend noting that the diagnoses provided correspond to larvae, perhaps it would be sufficient to state "Diagnosis (larva)", instead of "diagnosis of the larva" as it is repeated several times.

Line 280: TNT returns a score that is a modified number (not whole numbers) of steps when running implied weights. How was the tree length determined to be 301 if these were implied? Was the topology from the implied weights analysis used to constrain an equal weights analysis to produce a whole-value step number? I may be missing something, but if not, my concern is that the authors have not run the implied weights search correctly in TNT. Also, see the following:

The authors response to my initial comment: "The 4 trees obtained under implied weights (k=9.919, see following comment) are characterized by the same tree length of those resulting from the analysis under equal weights, so they are included among the 680 trees obtained enforcing the latter conditions. In this way, we selected the 4 most fitting tree among the 680 shorter trees. However, following the reviewer's suggestion, we included the strict consensus tree obtained under equal weight as supplementary information." This seems unlikely because 1) the length reported in the manuscript has changed from 305 to 301 steps – was this due to changing the k-value? If so, then the tree lengths are not the same for implied and equal weights trees. 2) Is it possible the authors have not sufficiently explored all possible tree space? This comment does not require an entire new round of reviews, in my opinion, unless the authors identify a source of error based on these questions.

Bootstrap and/or bremer support values must be made available for equal weights trees and bootstrap values for implied weights trees.

Given the general perception of ant lions in particular, it would be nice to include mention of ants as a prey item and the age and diversification of ants lying within the Cretaceous as well: e.g. Moreau, C.S., Bell, C.D., Vila, R., Archibald, S.B. and Pierce, N.E., 2006. Phylogeny of the ants: diversification in the age of angiosperms. *Science*, 312(5770), pp.101-104. Based on molecular data and Barden, P. and Grimaldi, D.A., 2016. Adaptive radiation in socially advanced stem-group ants from the Cretaceous. *Current Biology*, 26(4), pp.515-521. For fossils.

Responses to the reviewers' comments:

Reviewer #1 (Remarks to the Author):

I accept the authors' modifications. This is an important contribution to understand the earlier evolution of neuropteran larva. I recommend publication of the paper as it stands.

- We thank the reviewer for his/her careful attention and precious insights on the manuscript.

Reviewer #3 (Remarks to the Author):

Following up on my initial review, the authors have sufficiently responded to my comments and I look forward to seeing the manuscript published. A very nice set of paleontological descriptions coupled with phylogenetics. Below are my minor comments, however, I do not think that these should warrant a new round of reviews on their own, unless the phylogenetic questions inspire doubt in the authors claims.

- We really thank the reviewer for his/her valuable comments which greatly improved our manuscript.

*Line 36: "both stem- and crown-groups to extant families" should be "both stem- and crown-groups with respect to extant families" – not possible to be a crown group *to* modern families.*

- We agree with the reviewer and we changed the sentence. This was an oversight from our side.

Line 110: while I did recommend noting that the diagnoses provided correspond to larvae, perhaps it would be sufficient to state "Diagnosis (larva)", instead of "diagnosis of the larva" as it is repeated several times.

- We accepted the reviewer's suggestion.

Line 280: TNT returns a score that is a modified number (not whole numbers) of steps when running implied weights. How was the tree length determined to be 301 if these were implied?

- This means that the fittest trees were among the shortest trees, i.e., the algorithm did not find any fittest trees longer, which is to say that forcing the topology to some characters with fewer steps does not make others to increase the number of steps.

Was the topology from the implied weights analysis used to constrain an equal weights analysis to produce a whole-value step number?

- No, we did not apply any constrain to the analysis under equal weights. Having the fittest trees among the shorter ones is common when dealing with "relatively stable" backbone phylogenies, with only the more distal nodes varying (as is clear from the strict consensus under equal weights). See answer above.

I may be missing something, but if not, my concern is that the authors have not run the implied weights search correctly in TNT. Also, see the following:

The authors response to my initial comment: "The 4 trees obtained under implied weights ($k=9.919$, see following comment) are characterized by the same tree length of those resulting from the analysis under equal weights, so they are included among the 680 trees obtained enforcing the latter conditions. In this way, we selected the 4 most fitting tree among the 680 shorter trees. However, following the reviewer's suggestion, we included the strict consensus tree obtained under equal weight as supplementary information." This seems unlikely because 1) the length reported in the manuscript has changed from 305 to 301 steps – was this due to changing the k -value? If so, then the tree lengths are not the same for implied and equal weights trees. 2) Is it possible the authors have not sufficiently explored all possible tree space? This comment does not require an

entire new round of reviews, in my opinion, unless the authors identify a source of error based on these questions.

- We respectfully disagree with the reviewer's comment. The analyses under implied weights have been carried out correctly, the tree space have been explored sufficiently. However, the reviewer is right in his observation when says "*the length reported in the manuscript has changed from 305 to 301 steps*". In fact, differences with the previous trees are due to a few, minor, changes we made to the data matrix, as a consequence of a differing homology assessment suggested by comments provided during the first round of review.

Bootstrap and/or bremer support values must be made available for equal weights trees and bootstrap values for implied weights trees.

- We followed the reviewer's suggestion by adding branch supports to the trees obtained under equal weights and implied weights (see Supplementary note). However, instead of bootstrap supports, we preferred a different resampling method: jackknife resampling percentiles.

Given the general perception of ant lions in particular, it would be nice to include mention of ants as a prey item and the age and diversification of ants lying within the Cretaceous as well: e.g. Moreau, C.S., Bell, C.D., Vila, R., Archibald, S.B. and Pierce, N.E., 2006. Phylogeny of the ants: diversification in the age of angiosperms. Science, 312(5770), pp.101-104. Based on molecular data and Barden, P. and Grimaldi, D.A., 2016. Adaptive radiation in socially advanced stem-group ants from the Cretaceous. Current Biology, 26(4), pp.515-521. For fossils.

- We respectfully disagree with the reviewer's comments. Despite the vernacular name of these insects, which include the term "ant", antlion larvae (myrmeleontids) and other myrmeleontiforms are not myrmecophiles nor strictly associated with ants (Mansell, 1999 - Evolution and success of antlions (Neuropterida: Neuroptera: Myrmeleontidae). They are, instead, generalist predators feeding on any suitable insect. The peculiar vernacular name of these insects (of which there are equivalents in all European languages) has a mythological (not behavioural) origin. Moreover, myrmeleontiforms are not comparable to ants in ecology, behaviour and whole habits, thus a similar association is meaningless. Also of note is that according to recent time-divergence estimates (Wang, Y. et al. 2017 - Mitochondrial phylogenomics illuminates the evolutionary history of Neuropterida; Winterton, S. L. et al. 2018 - Evolution of Neuropterida and allied orders using anchored phylogenomics (Neuroptera, Megaloptera, Raphidioptera)), Myrmeleontiformia predate ants, branching off from the rest of the lacewing backbone already in the Triassic.